# T cell fate following *Salmonella* infection is determined by a STING-IRF1 signaling axis in mice

Sung-Moo Park [1,2], Tatsushi Omatsu [1,2], Yun Zhao[1], Naohiro Yoshida [1], Pankaj Shah[1], Rachid Zagani[1] & Hans-Christian Reinecker [1]*

The innate immune response following infection with entero-invasive bacterial species is triggered upon release of cyclic di-guanylate monophosphate (c-di-GMP) into the host cell cytosol. Bacterial c-di-GMP activates the intracellular Sensor Stimulator of Interferon Genes (STING), encoded by *Tmem173* in mice. Here we identify Interferon Regulatory Factor (IRF) 1 as a critical effector of STING-mediated microbial DNA sensing that is responsible for $T_H17$ cell generation in the mucosal immune system. We find that STING activation induces IRF1-dependent transcriptional programs in dendritic cells (DCs) that define T cell fate determination, including induction of Gasdermin D, IL-1 family member cytokines, and enzymes for eicosanoid synthesis. Our results show that IRF1-dependent transcriptional programs in DCs are a prerequisite for antigen-specific $T_H17$ subspecification in response to microbial c-di-GMP and *Salmonella typhimurium* infection. Our identification of a STING-IRF1 signaling axis for adaptive host defense control will aid further understanding of infectious disease mechanisms.

[1] Gastrointestinal Unit and Center for the Study of Inflammatory Bowel Disease, Massachusetts General Hospital, Harvard Medical School, Boston, MA 02114, USA. [2] These authors contributed equally: Sung-Moo Park, Tatsushi Omatsu. *email: hans-christian_reinecker@hms.harvard.edu

The regulated synthesis of cyclic di-guanylate monophosphate (c-di-GMP) plays a critical regulatory role in multiple species of bacteria[1]. C-di-GMP induces host defense responses through interaction with STING (also named *MITA*, *MYPS*, or *ERIS*, encoded by *Tmem173*)[2–4]. STING is a potent activator of the innate immune system that can recognize cytosolic dsDNA through distinct DNA sensors[5–11]. STING variants with single nucleotide polymorphisms differ in their ability to detect microbial-derived cyclic dinucleotide (CDN), such as c-di-GMP and cyclic deadenylate monophosphate (c-di-AMP)[12,13]. STING further is required for the detection of cytosolic DNA derived from nuclear and mitochondrial genomes[14,15] and the initiation of immune responses to DNA viruses and bacteria[10,16].

IRF1 was initially identified as inducer of the transcription of type I IFNs and IFN-inducible genes[17]. However, IRF1 mediates cell type-specific transcriptional programs that are key to host defenses initiated by a wide range of microbial pattern recognition receptors, such as Toll-like receptor (TLR), RIG-I like receptors, and cytokine receptors[18]. *Irf1*-deficient (*Irf1*$^{-/-}$) mice exhibit an impaired T cell subdifferentiation with enhanced $T_H2$ response due to defects in $T_H1$ differentiation[19,20]. Consequently, *Irf1*$^{-/-}$ mice are highly susceptible to bacterial and viral pathogens[20]. Conversely, *Irf1*$^{-/-}$ mice appear to be protected from autoimmune diseases including experimental autoimmune encephalomyelitis, Type II collagen-induced arthritis, and diabetes[21–23].

Mucosal DCs are heterogeneous and comprise several subtypes with different phenotypes and functional properties that sample luminal and circulatory antigens, and mediate host defenses against entero-invasive pathogens[24–26]. DCs can direct T-cell differentiation programs[27] leading to functionally distinct subsets of T helper cells, including $T_H1$, $T_H2$, $T_H9$, and $T_H17$ that are characterized by lineage-specific transcription factors and signature cytokines[28,29]. $T_H17$ cells that produce IL-17A, IL-17F, and IL-22 arise in the presence of transforming growth factor-beta (TGF-β) and IL-6[30,31]. $T_H17$ cells have an essential role in induction of protective immunity against bacterial and fungal pathogens but conversely can contribute to the pathogenesis of several autoimmune diseases[32–35].

Here we found that the recognition of c-di-GMP through STING in the mucosal immune system is a significant signal for the induction of $T_H17$ cell differentiation. STING activation by c-di-GMP initiated the migration of mucosal DCs and resulted in a significant increase in $T_H17$ cells in mesenteric lymph nodes (MLNs). We defined IRF1-dependent, IRF3-dependent, and IRF7-dependent transcriptional programs in DCs that controlled T cell differentiation in response to STING activation. We found that IRF1 was phosphorylated upon interaction with STING and was required for the induction of a specific transcriptional program that included Gasdermin D, IL-1 family cytokines, prostaglandin synthases as well as caspases. IL-1 and prostaglandins were able to compensate for the loss of IRF1 and re-establish and sustained $T_H17$ polarization during antigen-specific T cell activation. Collectively, we identified the STING–IRF1 signaling axis as essential for the control of mucosal $T_H17$ cell responses that signify protective host defense activation against entero-invasive pathogen such as *S. typhimurium*.

## Results

**STING signaling activates dendritic cells and induces $T_H17$ cells.** We carried out intranasal immunizations with ovalbumin (OVA) in the absence or presence of c-di-GMP in wild-type C57BL/6 mice to determine whether c-di-GMP can enhance antigen-specific T cell activation in the mucosa-associated immune system. After three immunizations over a 7-day period cervical lymph node (CLN) and MLN cells were isolated and re-stimulated

with OVA for 48 h. As demonstrated in Fig. 1a, c-di-GMP and OVA together were able to induce significant increase in IL-17A and IFN-γ secretion in CLNs as well as in MLNs compared to OVA alone. The induction of IL-17A by c-di-GMP required STING signaling as intranasal immunizations in *Tmem173*$^{-/-}$;*Il17a*-GFP reporter mice resulted in significantly less IL-17A-expressing T cells in CLNs and MLNs compared to wild-type mice (Fig. 1b). To determine whether STING signaling was able to activate mucosal DCs, we crossed *Tmem173*$^{-/-}$ mice into the CAG::KikGR (KikGR) background to be able to track the migration of lamina propria (LP)-DCs after photoconversion in response to c-di-GMP. Photo-converted red-fluorescent immune cells that originated in the small intestine were found in MLNs of KikGR mice 18 h after exposure (Fig. 1c). Stimulation with c-di-GMP induced a significantly enhanced migration of SI-LP CD11c$^+$MHCII$^+$ dendritic cells to MLNs in KikGR but not in KikGR;*Tmem173*$^{-/-}$ mice. Over an 18 h period, c-di-GMP increased the percentage of mucosa-derived DC that reached the MLN from 4.7% to 11.7% (Fig. 1c). We determined the migration of the two classical CD103$^+$CD11b$^+$ DCs and CD103$^+$CD11b$^-$ DCs subsets in wild-type mice (Fig. 1c). Both subsets migrated in response to c-di-GMP stimulation, with lamina propria MHCII$^+$CD103$^+$CD11b$^+$DCs increasing 3.6-fold and MHCII$^+$CD103$^+$CD11b$^-$ DCs by 4-fold in MLNs (Fig. 1c). The two major MHC class II (MHC II)-positive myeloid-derived cell types in the SI-LP that could be responsible for $T_H17$ induction in the LP are Zbtb46$^+$CD103$^+$CD11b$^+$ DCs or Cx3cr1$^+$CD103$^-$CD11b$^+$ macrophage-derived DCs (Fig. 1d). Both Zbtb46$^+$CD103$^+$CD11b$^+$ DCs and Cx3cr1$^+$CD103$^-$CD11b$^+$ myeloid cells increased their IL17A stimulatory capacity upon c-di-GMP stimulation by 1.5 and 3.5 fold, respectively (Fig. 1e). However, we found that Zbtb46$^+$CD103$^+$CD11b$^+$ DCs induced significantly (229%) more *Il17a* expression in OT-II transgenic T cells compared to Cx3cr1$^+$CD103$^-$CD11b$^+$ DCs (Fig. 1e). Furthermore, stimulation of mucosal Zbtb46$^+$ DC with c-di-GMP resulted in the significant increase in *IFNb1*, *Il12b*, *IL23a*, and *Il6* mRNA expression (Fig. 1f) all of which have been linked to the induction of $T_H17$ cells. Taken together, microbial c-di-GMP-activated innate immune responses in Zbtb46$^+$CD103$^+$CD11b$^+$ and Cx3cr1$^+$CD103$^-$CD11b$^+$ LP-DCs for the induction $T_H17$ cells in the mucosal immune system.

**STING induces phosphorylation and nuclear translocation of IRF1.** RNA-sequencing-based expression analysis of genes encoding for IRF-type transcription factors revealed that c-di-GMP induced the transcription of *Irf1*, *Irf7*, *Irf8*, and *Irf9* in DCs (Fig. 2a). We found a rapid and significant induction of *Irf1* within 4 h of stimulation with c-di-GMP that was sustained for 18 h and required STING expression, while *Irf7* expression significantly increased over 18 h after c-di-GMP stimulation (Fig. 2b). In contrast, expression of mRNA for *Irf3* remained unchanged in wild-type and *Tmem173*$^{-/-}$ BMDCs at 4 h, and was significantly reduced after 18 h after c-di-GMP stimulation in *Tmem173*$^{-/-}$ BMDCs (Fig. 2b).

STING signaling by c-di-GMP not only induced *Irf1* gene expression but may also result in the sustained nuclear recruitment of IRF1 that was absent in *Tmem173*$^{-/-}$ DCs (Fig. 2c). Co-immunoprecipitation experiments revealed that the human H232R and R232H STING variants formed protein complexes that contained IRF1 and, conversely, IRF1-specific antibodies were able to pull down protein complexes that contained the STING variants (Fig. 2d). Furthermore, STING and IRF1 aggregated in the same signaling complexes in primary DCs, as anti-IRF1, but not control IgG, pulled down immunocomplexes that contained STING (Fig. 2e). To determine whether STING induced IRF1 phosphorylation, we co-transfected HA-

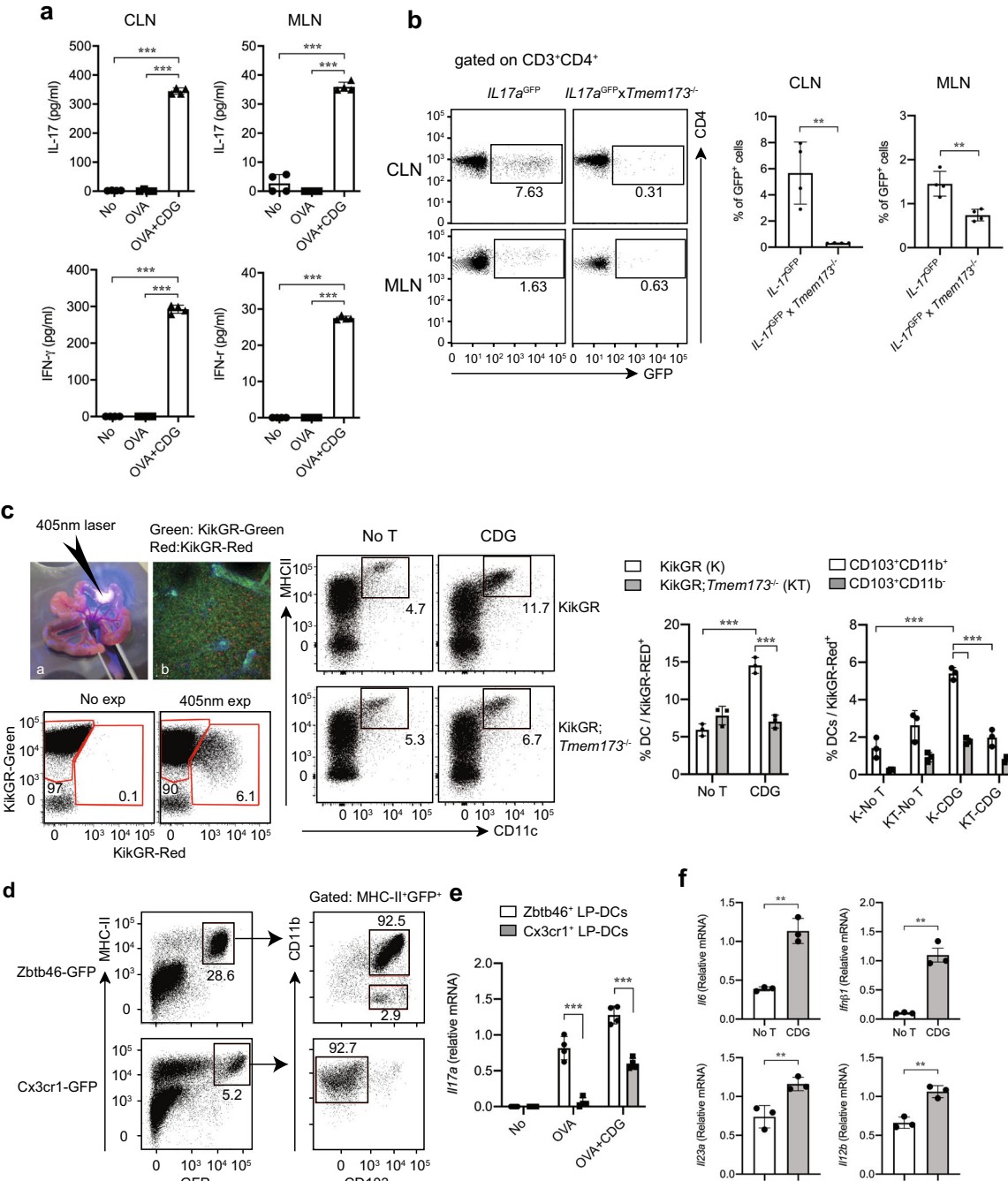

**Fig. 1 STING signaling in mucosal DCs induces T$_H$17 cells. a**, **b** Cytokines expression by CLN and MLN cells obtained from wild-type mice at 7 days after the last immunization with 20 μg OVA without or with 25 μg c-di-GMP. **a** Cells were cultured for 48 h ex vivo with 20 μg/ml OVA, and cytokines were measured by ELISA. $n = 4$. ***$p < 0.001$. **b** Representative data (left) depicting the frequency (%) of IL-17A-GFP$^+$ CD4$^+$ cells from *Il17a*-GFP mice or *Tmem173*$^{-/-}$;*Il17a*-GFP mice. $n = 4$. **$p < 0.01$. **c** Analysis of DCs migration from SI to MLNs upon c-di-GMP stimulation. Small intestines (SI) of KikGR mice and KikGR;*Tmem173*$^{-/-}$ mice were exposed with a 405 nm laser and i.p. injected with c-di-GMP (200 μg/mouse) 18 h before sacrifice. Migrated cells were defined in MLN by gating CD45$^+$ KikGR-Red$^+$ cells using flow cytometry. Graph (left) shows the percentage of CD11c$^+$ MHCII$^+$ DC in CD45$^+$ KikGR-Red$^+$ migrated cells. Graph (right) shows the percentage of CD103$^+$CD11b$^+$ or CD103$^+$CD11b$^-$ DCs in CD11c$^+$MHCII$^+$ KikGR-Red$^+$ migrated DCs. $n = 3$. ***$p < 0.001$. **d** Analysis of MHCII$^+$ Zbtb46$^+$ or Cx3cr1$^+$ LP-DCs. LP-DCs were isolated from Zbtb46-GFP or Cx3cr1-GFP mice and analyzed for CD103 and CD11b expressions. **e** Analysis of *Il17a* mRNA expression after antigen-specific T cell activation by Zbtb46$^+$ or Cx3cr1$^+$ LP-DCs. Zbtb46-GFP and Cx3cr1-GFP mice were injected (i.p.) with c-di-GMP (200 μg/mouse) on −5, −3 and −1 day prior to sacrifice. CD11c$^+$ GFP$^+$ DCs were sorted from SI-LP and incubated with naive T cells isolated from spleen of OT-II transgenic mice with OVA (50 μg/ml) for 5 days (DC:T cell = 1:5). $n = 4$. ***$p < 0.001$. **f** mRNA expression was analyzed by qRT-PCR in Zbtb46$^+$ LP-DCs. CD11c$^+$ Zbtb46-GFP$^+$ cells were sorted by flow cytometry and incubated with c-di-GMP (50 μg/ml) for 4 h. $n = 3$, **$p < 0.01$. *p*-values, one-way **a**, two-way **c**, **e** ANOVA followed by Tukey's multiple-comparisons test or two-tailed unpaired Student's *t*-test. **b**, **f** Data are representative of two independent experiments **a**–**c** or are from three indpendent experiments **d**–**f**.

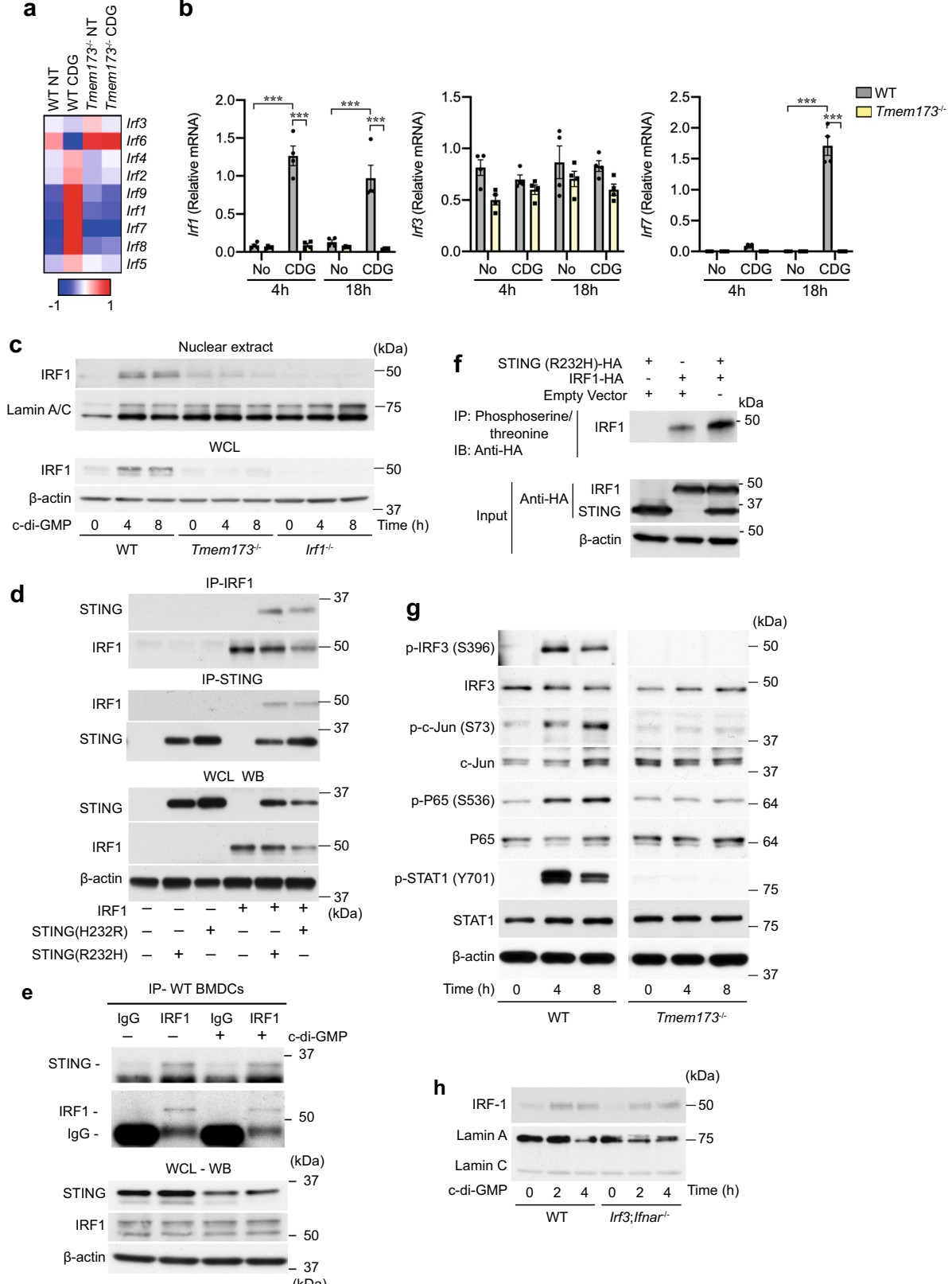

STING and HA-IRF1 plasmids into HEK293 T cells and pulled down immunocomplexes using anti-phosphoserine/threonine antibody. We found that in the presence of STING the amount of phosphorylated IRF1 increased compared to IRF1 expression alone (Fig. 2f).

The nuclear recruitment of IRF1 occurred in the context of additional signaling events activated by c-di-GMP including the phosphorylation of IRF3, c-Jun, NF-κB p65, and STAT1, all of which were activated dependent on the presence of STING (Fig. 2g). However, the nuclear recruitment of IRF1 occurred

**Fig. 2 IRF1 is activated upon interaction with STING. a** Differential gene expression analysis of IRFs in wild-type and $Tmem173^{-/-}$ BM-DCs after stimulation by c-di-GMP (50 µg/ml) for 18 h. **b** Real-time PCR analysis of *Irf1*, *Irf3*, and *Irf7* mRNA expressions in BM-DCs after stimulation with c-di-GMP for 4 and 18 h. Results are presented relative to normalized expression of the 18S ribosomal RNA. $n = 4$. ***$p < 0.001$. (Two-way ANOVA followed by Tukey's multiple-comparisons test.) **c** BMDCs from wild-type, $Tmem173^{-/-}$, and $Irf1^{-/-}$ ($1 \times 10^6$ per time point) were stimulated with 50 µg/ml of c-di-GMP as indicated. Cells were lysed and nuclear compartments were analyzed by Western blotting with indicated antibodies (Abs). IRF-1 was visualized first; the membrane was then stripped and re-probed with anti-Lamin A/C. **d** Human STING variants and IRF1 were co-expressed in HEK239T cells and either STING or IRF1 immunoprecipitated, and detected with specific antibodies by Western blotting. Expression levels of STING or IRF1 were validated in total cell extracts by SDS–PAGE and immunoblotting. **e** Immunoprecipitation of IRF1 from primary wild-type BMDCs that were stimulated with 25 µg/ml c-di-GMP. Anti-STING was used to detect proteins by Western blotting. **f** Antibodies detecting phosphoserine and/threonine were used to precipitate IRF1 from HEK293T cells that were co-transfected with HA-STING and HA-IRF1 encoding plasmids. IRF1 and STING were detected using HA antibody. **g** BMDCs from wild-type and $Tmem173^{-/-}$ ($1 \times 10^6$ cells per time point) were stimulated with 50 µg/ml of c-di-GMP as indicated. Cells were lysed and analyzed by Western blotting with indicated antibodies (Abs). p-IRF-3 p-STAT1, p-c-Jun, p-p65 were detected first and the membranes stripped and re-probed with anti-IRF-3, anti-STAT1, anti-c-Jun, anti-p65, and anti-β-actin. One of two independent experiments is shown. **h** Immunoblot analysis of the nuclear recruitment of IRF1 in BMDCs isolated from wild-type or $Irf3;Ifnar^{-/-}$ mice upon STING activation Data are representative of two independent experiments **c**–**h** or are from three independent experiments **b**.

---

independent of IRF3 and interferon-α/β receptor (IFNAR) signaling in $Irf3;fnar^{-/-}$ BMDCs (Fig. 2h). Together these data demonstrated that IRF1 can be activated by STING signaling initiated by c-di-GMP in DCs.

**IRF1 mediates STING-dependent $T_H17$ cell polarization.** To gain insight into the role of IRF1 for STING-dependent induction of $T_H17$ cells, we isolated $Irf1^{-/-}$, $Irf3/7^{-/-}$ BMDCs and determined their ability to induce antigen-specific T cell polarization. We discovered that $Irf1^{-/-}$ BMDCs were unable to induce IL-17A production in naïve OT-II T cells in contrast to wild-type DCs during antigen presentation in the presence of c-di-GMP (Fig. 3a). Furthermore, c-di-GMP stimulation of $Irf1^{-/-}$ BMDCs also resulted in significantly less IFN-γ expression during antigen recognition (Fig. 3a). $Irf3/7^{-/-}$ BMDCs were also impaired in their ability to induce IL-17A production in T cells after c-di-GMP stimulation but were able to induce IFN-γ levels in T cells that were comparable to those observed wild-type BMDCs (Fig. 3b). In contrast, $Irf7^{-/-}$ BMDCs were able to induce IL-17A and IFN-γ during antigen-specific T cell activation after c-di-GMP stimulation comparable to wild-type BMDCs (Fig. 3c).

To determine whether IRF1 was required for the generation of $T_H17$ cell in vivo, CellTrace$^{TM}$ Violet (Molecular Probes)-labeled OT-II Naïve T cells were transferred into wild-type and $Irf1^{-/-}$ mice i.v. and CD4$^+$ T cell proliferation measured 5 days after intranasal immunization by flow cytometry (Fig. 3d). The proliferation of adoptively transferred OT-II cells was significantly impaired in $Irf1^{-/-}$ mice compared to wild-type mice (Fig. 3d). Furthermore, FACS-sorted labeled OT-II cells isolated from $Irf1^{-/-}$ mice lacked detectable *Il17a* and *Ifnγ* mRNA expressions compared to T cell isolated from wild-type mice 5 days after intranasal immunizations (Fig. 3e). We also carried out intra-peritoneal (i.p.) immunizations with OVA and c-di-GMP, and analyzed the resulting induction of *Il17a* and *Ifnγ* in CD4$^+$ T cells isolated from MLNs (Fig. 3f). Both STING and IRF1-deficient mice induced significantly less mRNA encoding for IL-17A and IFN-γ in CD4$^+$ T cells in MLNs (Fig. 3f).

We next investigated whether STING-signaling induced innate immune stimuli that create $T_H17$-polarizing micro-environments by BMDCs from wild-type (C57BL/6), $Tmem173^{-/-}$, $Irf1^{-/-}$, $Irf7^{-/-}$, and $Irf3/7^{-/-}$ mice. We found that $Tmem173^{-/-}$ BMDCs were unable to respond to c-di-GMP stimulation with the expression of T-cell-polarizing cytokines (Fig. 3g). Surprisingly, $Irf1^{-/-}$, $Irf3/7^{-/-}$, and $Irf7^{-/-}$ BMDCs expressed significantly more *Il6* and *Il23a* after c-di-GMP stimulation compared to wild-type BMDCs. In addition, *Il12a* expression was significantly reduced in $Irf1^{-/-}$ BMDCs during STING activation (Fig. 3g).

Further, $Irf1^{-/-}$ and $Irf3/7^{-/-}$ DCs expressed significantly less *Il27* mRNA compared to wild-type and $Irf7^{-/-}$ DCs (Fig. 3g).

Together, these experiments showed that c-di-GMP directly induced the activation of IRF1 in DCs to allow the differentiation of $T_H17$ cells during antigen-specific T cell activation in vitro and in vivo. However, the data also indicated that IRF1 and IRF3 contributed distinct signals to $T_H17$-cell differentiation that critically functioned upstream of IL-6 and IL-23.

**Microbial CDNs induce a IRF1-specific transcriptional program in DCs.** To more precisely identify the transcriptional programs by which STING signaling enables DC to induce $T_H17$ cells, we performed RNA-seq analysis of BMDCs from wild-type, $Tmem173^{-/-}$, $Irf1^{-/-}$, and $Irf3/7^{-/-}$ mice after 18 h of stimulation with c-di-GMP. Using Cuffdiff analysis, a total of 4588 genes were found to be differentially expressed by more than two-fold and a false discovery rate (FDR) cutoff of 0.05 between indicated groups (Supplementary Data 1). The Venn diagram in Fig. 4a represents the overlap of up-regulated or down-regulated genes comparing wild-type responses to c-di-GMP with those found in $Irf1^{-/-}$, $Irf3/7^{-/-}$, and $Tmem173^{-/-}$ mice demonstrating that STING together with *Irf1* and *Irf3/7* conveyed the majority of the transcriptional response to c-di-GMP.

Hierarchical-clustering analysis was used to distinguish genes that were induced by c-di-GMP stimulation in wild-type BMDCs but not by $Tmem173^{-/-}$ BMDCs (Fig. 4b). 1869 differentially expressed genes that were significantly upregulated more than two-fold could further be differentiated into clusters that were dependent on *Irf1* and/or *Irf3/7* distinguishing six signatures that were dependent on *Irf1* or *Irf3/7* alone or regulated by both *Irf1* and *Irf3/7* together (Supplementary Data 2, Fig. 4b). We were able to identify specific clusters of co-regulated genes that were specifically dependent on *Irf1* (clusters 2, 6). We also identified programs that were dependent on Irf3/7 alone (clusters 4 and 5) or required *Irf1* and *Irf3/7* (clusters 1, 3). In addition, genes in cluster 3 were significantly elevated in response to c-di-GMP in $Irf1^{-/-}$ and $Irf3/7^{-/-}$ BMDCs with a subgroup of genes specifically further elevated in $Irf3/7^{-/-}$ BMDCs. To define biological relevance of each cluster, we used gene sets from pathway databases as available in the PathCards and Molecular Signature Database (MSigDB)[36,37]. Gene set enrichment analysis (GSEA) in MSigDB data revealed that differently regulated genes (genes in clusters 2 and 6) by $Irf1^{-/-}$ BMDCs compared with both wild-type and $Irf3/7^{-/-}$ BMDCs are affiliated with IFN-γ, IFN-α, TGF-β, and IL6-JAK-STAT3 signaling gene signatures (Fig. 4c).

IRF1-dependent cluster 2 contained Caspases (*Casp4* and *Casp12*), IL-1-signaling-associated genes (*Il1α*, *Il1β*, *Il1rn*, *Fos*,

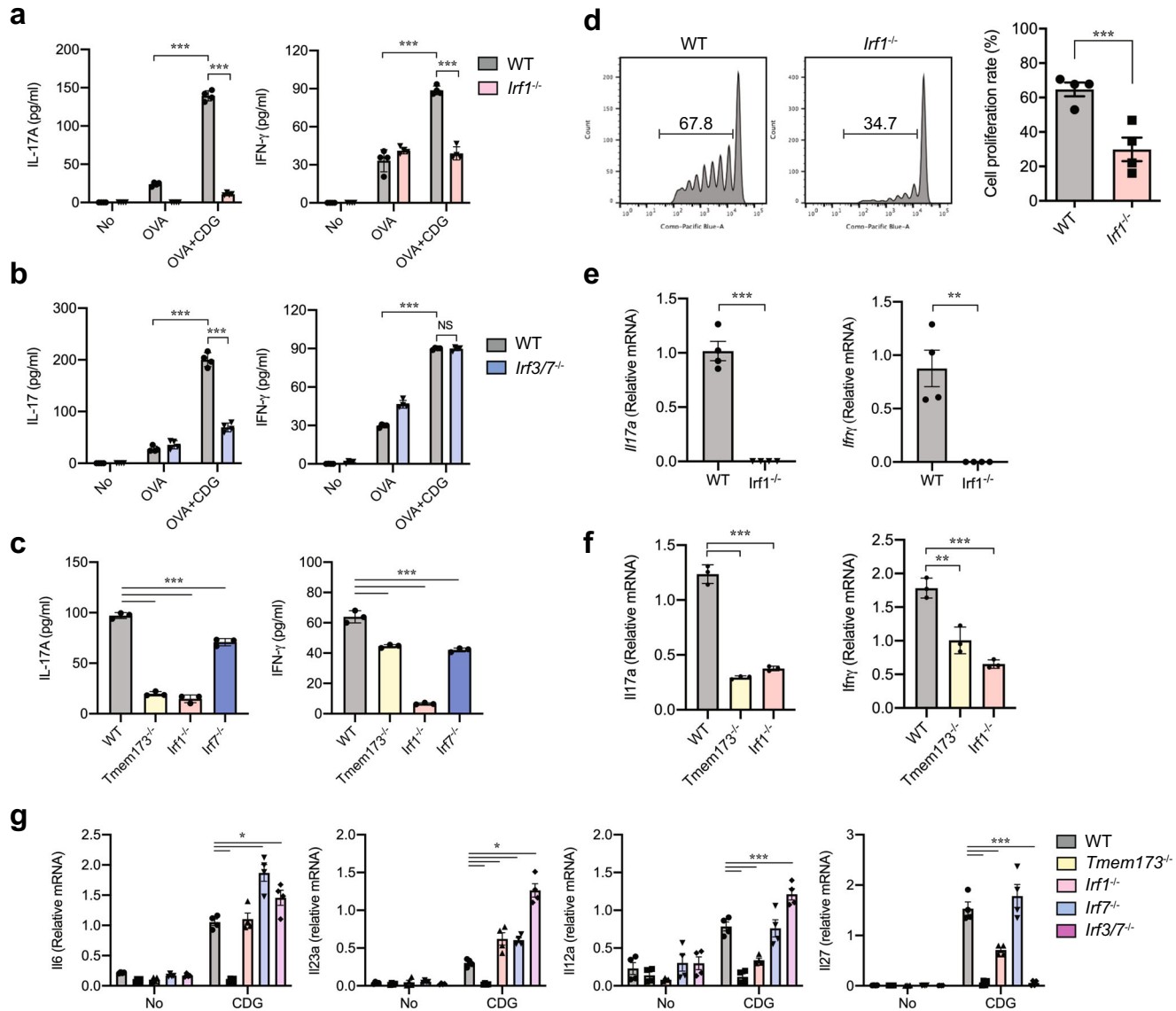

**Fig. 3 IRF1 expression in DCs controls antigen-specific T_H17 differentiation. a–c** $2 \times 10^4$ BMDCs from wild-type, $Irf1^{-/-}$, $Irf3/7^{-/-}$, and $Irf7^{-/-}$ mice were stimulated with PBS, 20 µg/ml of OVA or 20 µg/ml of OVA plus 25 µg/ml of c-di-GMP for 18 h and co-cultured for 4 days with $1 \times 10^5$ MACS-sorted naïve OT-II T cells. Production of IL-17A and IFN-γ was determined in supernatant of BMDC from **a** wild-type or $Irf1^{-/-}$ or **b** $Irf3/7^{-/-}$ mice co-cultured naïve OT-II cells. $n = 4$. ***$p < 0.001$. **c** Measurement of IL-17A and IFN-γ in supernatant of BMDC from wild-type, $Irf1^{-/-}$, $Irf3/7^{-/-}$, and $Irf7^{-/-}$ mice co-cultured naïve OT-II cells. $n = 3$. ***$p < 0.001$. **d** and **e** wild-type ($n = 4$) and $Irf1^{-/-}$ ($n = 4$) mice were (i.v.) injected $2 \times 10^6$ Cell trace Violet labeled naïve OT-II cells on day 0. On day 1, the mice were immunized with OVA plus c-di-GMP (i.n.). **d** Cell proliferation analysis in adoptively transferred OT-II cells from wild-type and $Irf1^{-/-}$ mice 5 days after immunization in cervical lymph nodes. $n = 4$. ***$p < 0.001$. **e** $Il17a$ and $Ifn\gamma$ mRNA expression was quantified by qRT-PCR in sorted Cell trace Violet positive OT-II cells after 5 days of immunization. $n = 4$, ***$p < 0.001$. **f** qRT-PCR analysis of $Il17a$ and $Ifn\gamma$ gene expressions in sorted CD4+ T cells in MLN from wild-type, $Tmem173^{-/-}$, and $Irf1^{-/-}$ mice immunized with OVA plus c-di-GMP through i.p. route. $n = 3$. **$p < 0.01$ and ***$p < 0.001$. **g** qRT-PCR analysis of the indicated genes in BMDCs from wild-type, $Tmem173^{-/-}$, $Irf1^{-/-}$, $Irf7^{-/-}$, and $Irf3/7^{-/-}$ mice after 18 h stimulation by 50 µg of c-di-GMP. $n = 4$. *$p < 0.05$, ***$p < 0.001$. $p$-values, one-way **c**, **f**, two-way **a**, **b**, **g** ANOVA followed by Tukey's multiple-comparisons test or unpaired two-tailed Student's $t$-test **d**, **e**. Data are representative of two **d**–**f** or three **a**, **b**, **c**, **g** independent experiments.

$Nod1$, and $Nod2$) and genes responsible for prostaglandin synthesis and regulation ($Ptgs2$, $Ptger3$, $Ptges$, $Ptgis$, $Cyp11a1$, $Edn1$, $Pla2g4a$, $Akr1a1$, and $Prdx1$) all off which failed to upregulate in $Irf1^{-/-}$ BMDCs upon c-di-GMP stimulation (Fig. 4d). The expression of Gasdermin D ($Gsdmd$), noted as effector of pyroptosis and formation of pore for releasing IL-1β, $Ptger2$, encoding for the Prostaglandin E2 (PGE2) receptor EP2, and Caspases ($Casp1$ and $Casp7$) were significantly impaired in both $Irf1^{-/-}$ and $Irf3/7^{-/-}$ BMDCs upon c-di-GMP stimulation

(Fig. 4f). The expression of $Il27$, $Maf$, and $Il12rb1$ recognized as IL-27-T_H1-axis-associated genes were significantly reduced in $Irf1^{-/-}$ BMDCs, while expression of these genes was significantly enhanced in $Irf3/7^{-/-}$ BMDCs after c-di-GMP stimulation (Fig. 4d). Genes associated with TGF-β signaling ($Smurf1$, $Fkbp1a$, $Pmepa1$, $E2f5$, $Cdkn2b$, and $Skil$), IFN-α/β signaling ($Socs3$, $Ifna4$, $Tyk2$, $Isg15$, $Irf4$, and $Adar$) and CTLA-4 inhibitory signaling ($Yes1$, $PPP2R5B$, $CD80$, and $CD86$) associated genes were increased in $Irf1^{-/-}$ BMDCs upon c-di-GMP stimulation

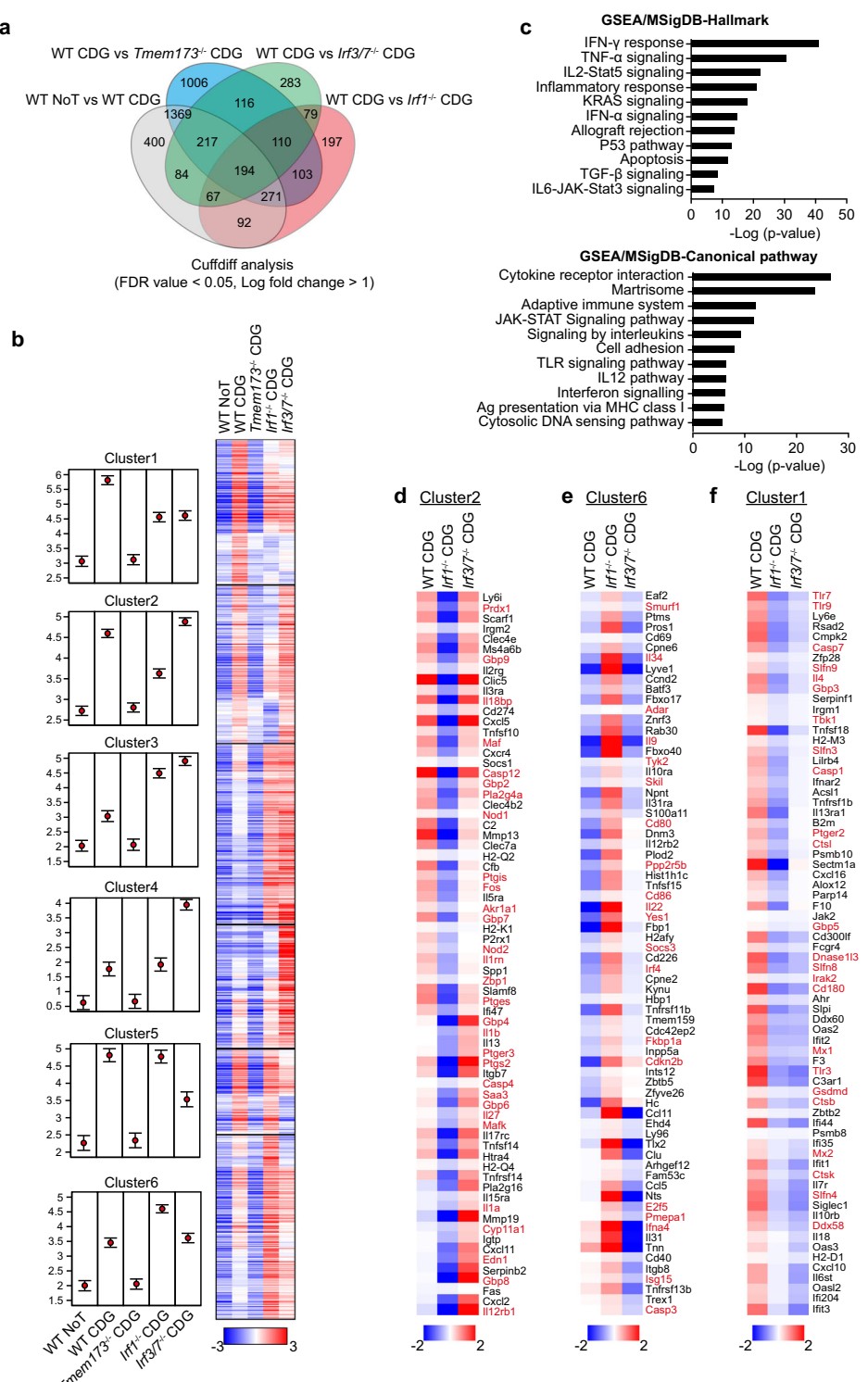

**Fig. 4 IRF1 and IRF3/7 control unique gene expression signatures upon STING activation.** RNA-seq analyses of BMDCs from wild-type, *Tmem173⁻/⁻*, *Irf1⁻/⁻*, and *Irf3/7⁻/⁻* mice 18 h after 50 μg/ml of c-di-GMP stimulation. **a** Venn diagram of at least two-fold differentially expressed genes (DEGs) in the indicated pairwise cuffdiff analysis from wild-type, *Tmem173⁻/⁻*, *Irf1⁻/⁻*, or *Irf3/7⁻/⁻* BMDCs. DEGs were identified using an FDR cutoff <0.05 and a fold change cutoff >2. **b** GSEA/MisDB analysis showing significant pathways in "Hallmark" and "Canonical pathway" categories. **c** Starwars plot generated in Seqmonk showing the expression of DEG signatures in CDG-dependent gene clusters in wild-type, *Tmem173⁻/⁻*, *Irf1⁻/⁻*, or *Irf3/7⁻/⁻* BMDCs. The individual data points are given in the supplemental data sets for the identified clusters (Supplementary Data 1). For each group the mean quantification of gene signatures is represented by a filled circle and the confidence interval (standard error) presented by error bars. The Heat map shows the hierarchical clustering of DEGs from wild-type, *Tmem173⁻/⁻*, *Irf1⁻/⁻*, and *Irf3/7⁻/⁻* BMDCs stimulated for 18 h by c-di-GMP. **d–f** Heat maps showing genes contained in the indicated clusters in **c** that contain IRF1 and IRF3/7-dependent gene signatures with key regulators high-lighted in red.

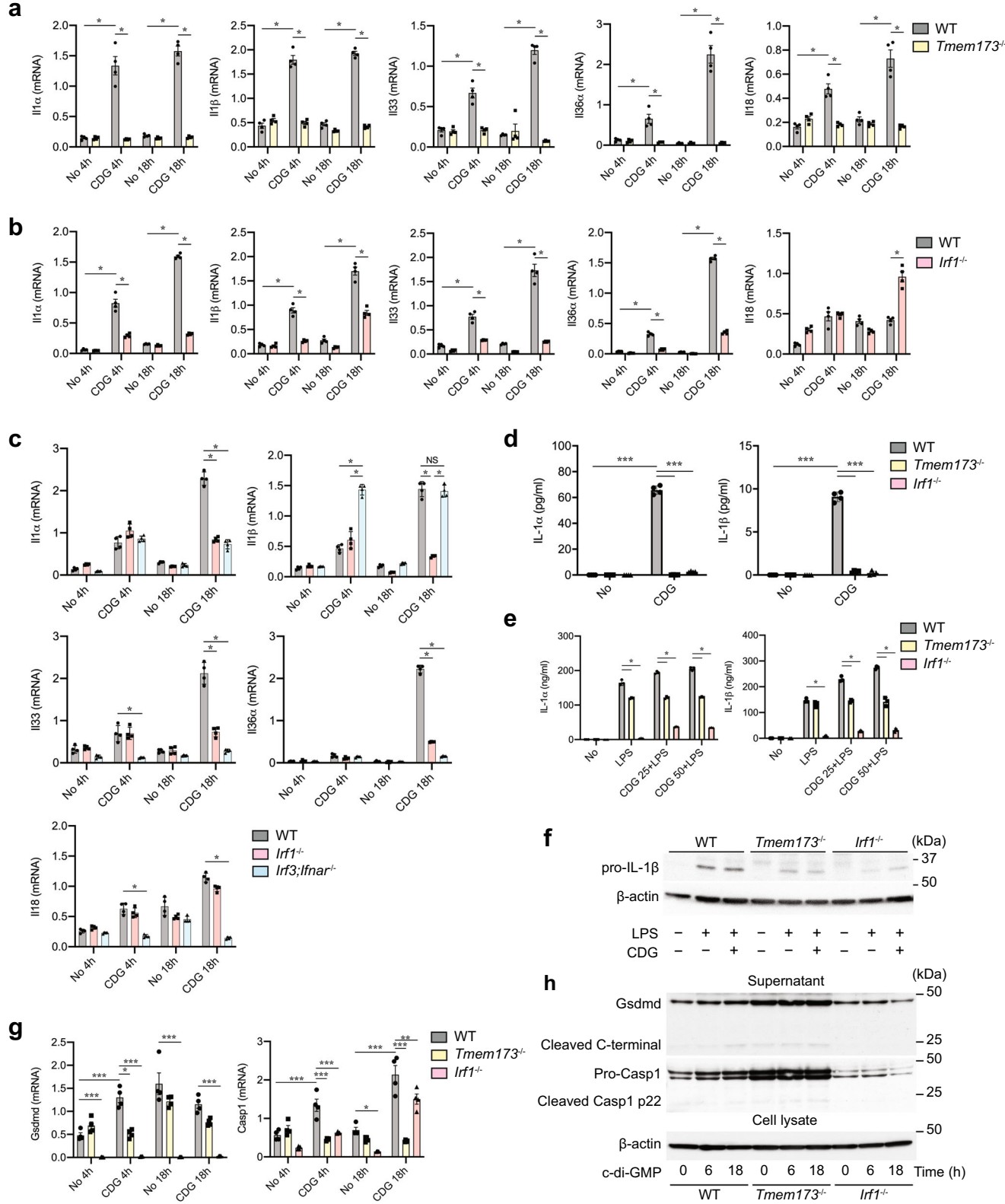

(Fig. 4e). In Cluster 3, *Il23a*, *Il23r*, *Il6*, and *Ebi3* were among genes significantly higher expressed in *Irf1*[−/−] and *Irf3/7*[−/−] BMDCs compared to wild-type BMDCs after c-di-GMP stimulation (Supplementary Fig. 3a). In contrast, TLR pathway genes (*Tlr3*, *Tlr7*, *Tlr9*, *CD180*, *Tbk1*, *Irak2*, *Ctsb*, *Ctsk,* and *Ctsl*) were significantly reduced in both *Irf1*[−/−] and *Irf3/7*[−/−] BMDCs upon c-di-GMP stimulation (Fig. 4f). Expression levels of IFN-γ

production-associated genes (*Il12a*, *Il12b*, *IL10*, *Ripk2,* and *Tnf*) were significantly up-regulated in *Irf3/7*[−/−] BMDCs (Supplementary Fig. 3b). RIG-1/MDA5 mediated induction of IFN-α/β pathway genes (*Trim25*, *Ifih1*, *Rnf135*, *Dhx58*, and *Irf7*), Notch pathway genes (*Src*, *Ccnd1*, *Hdac10*, and *Kat2b*), Caspases (*Casp3* and *Casp6*) were significantly down-regulated in *Irf3/7*[−/−] BMDCs (Supplementary Fig. 3c).

**Fig. 5 IRF1 is required for T$_H$17 cell differentiation during STING activation. a** qRT-PCR analysis of the Il1 family genes in BMDCs from Tmem173$^{-/-}$ versus wild-type mice after 4 and 18 h stimulation by 25 μg/ml of c-di-GMP. n = 4. *p < 0.001. **b** mRNA expression of Il1 family genes in BMDC from Irf1$^{-/-}$ versus wild-type mice after stimulation with c-di-GMP at indicated time. n = 4. *p < 0.001. **c** qRT-PCR analysis of the indicated genes in BMDCs from wild-type, Irf1$^{-/-}$, and Irf3;Ifnar$^{-/-}$ mice after 4 and 18 h stimulation by 25 μg/ml of c-di-GMP. n = 4. *p < 0.001. **d** Production of IL-1α and IL-1β in BMDCs from Irf1$^{-/-}$, Tmem173$^{-/-}$ versus wild-type mice after 18 h c-di-GMP stimulation. n = 4, *p < 0.001. **e, f** wild-type, Tmem173$^{-/-}$ and Irf1$^{-/-}$ BMDCs were LPS-primed for 3 h followed by stimulation with c-di-GMP for 6 h and supernatant were measured for IL-1α and IL-1β by ELISA **e**. n = 3. *p < 0.001. **f** Immunoblot analysis of IL-1β in cell lysates. **g** qRT-PCR analysis of Casp1 and Gsdmd gene expressions in BMDCs from wild-type, Tmem173$^{-/-}$, and Irf1$^{-/-}$ mice after 4 and 18 h stimulation by 25 μg/ml of c-di-GMP. n = 4 *p < 0.05, ***p < 0.001. **h** Immunoblot analysis of Gasdermin D and Caspase-1 in cell supernatant of BMDCs from wild-type, Tmem173$^{-/-}$, and Irf1$^{-/-}$ mice after 6 and 18 h c-di GMP stimulation. P-values, one-way **h**, two-way ANOVA **a**, **b**, **c**, **d**, **e**, and **g** followed by Tukey's multiple-comparisons test. Data shown are the mean value ± SD in ELISA or mean value ± SEM in qRT-PCR experiments. Data are representative of two independent experiments with similar results.

Collectively, this data identified a IRF1-specific genes expression signature characterized by IL-1 family cytokines, Gasdermin D and eicosanoid enzymes that was induced in DCs during the recognition of microbial c-di-GMP by STING.

**IRF1 mediates the activation of the IL-1/Gasdermin D system.** To confirm the RNA-sequencing findings, we carried out independent experiments to define mRNA and protein expression of IL-1 family cytokines, Gasdermin D and caspase activation in response to c-di GMP stimulation of DCs. Indeed. Tmem173$^{-/-}$ DCs expressed significantly less mRNA encoding for IL-1α, IL-1β, IL-33, IL-36α, and IL-18 for 4 and 18 h after stimulation with c-di-GMP (Fig. 5a). We found that IRF1 was required for the induction of Il1α and Il1β mRNA expressions and protein secretion by DCs (Fig. 5b, d). Irf1$^{-/-}$ DCs expressed significant less Il1α and Il1β mRNA expressions after 4 and 18 h of stimulation with c-di-GMP compared to wild-type DCs (Fig. 5b). Consequently, Irf1$^{-/-}$ as well as Tmem173$^{-/-}$ DCs failed to secrete significant amounts of IL-1α and IL-1β when compared to wild-type DCs upon STING activation for 18 h (Fig. 5d).

IRF1 was specifically required for the induction of Il1β and Il12b by c-di-GMP as induction of these cytokines was not impaired in Irf3;Ifnar double-deficient DCs (Fig. 5c and Supplementary Fig. 1d). In contrast, Irf1$^{-/-}$ and Irf3$^{-/-}$;Ifnar$^{-/-}$ DCs expressed significantly less Il1α, Il33, and Il36α mRNA expressions, suggesting that either IRF3 or response through the IFNAR downstream of IRF1 contributed to the induction of these cytokines (Fig. 5c). Furthermore, the expression of Il12rb1 and Il12rb2 in response to c-di-GMP stimulation was significantly reduced in Irf1$^{-/-}$ DCs compared to wild-type DCs (Supplementary Fig. 1a). IRF1 and IRF3/IFNAR together were also required for the control of Ptges1, Ptgs2, Gbp10, Gbp6, Il23r, and Ifnα2 mRNA expression by c-di-GMP (Supplementary Fig. 1b). However, Il18 mRNA was induced by c-di-GMP independent on IRF1 but was significantly less induced in Irf3$^{-/-}$;Ifnar$^{-/-}$ DCs compared to wild-type DCs (Fig. 5c). IRF3 and IFNAR together but not IRF1 were also required for the induction of classical type I interferon response genes, such as Mx2, Slfn5, and Ifnβ (Supplementary Fig. 1c). Further, neither IRF3 nor IFNAR was required for the regulation of Tgfβ1 mRNA expression by c-di-GMP (Supplementary Fig. 1d).

To determine whether IRF1 was a master regulator for IL-1α and IL-1β expressions we primed wild-type, Irf1$^{-/-}$, and Tmem173$^{-/-}$ DCs with LPS for 3 h before stimulation with c-di-GMP for additional 6 h. LPS and c-di-GMP demonstrated synergistic induction of IL-1α and IL-1β protein secretion (Fig. 5e). Remarkably, we found that IRF1 was also required for the induction of IL-1α and IL-1β by LPS (Fig. 5e). In contrast, Tmem173$^{-/-}$ DC were able to respond to LPS with reduced induction of IL-1α and IL-1β (Fig. 5e). Western blot analysis of DCs lysates demonstrated that Irf1$^{-/-}$ DCs produced less un-

cleaved IL-1β compared to wild-type and Tmem173$^{-/-}$ DCs (Fig. 5f). Importantly, direct inflammasome activation by a combination of ATP and LPS was not impaired in Tmem173$^{-/-}$ or Irf1$^{-/-}$ DCs (Supplementary Fig. 2).

Important for the secretion of IL-1β is the activation of Caspase-1 and Gasdermin D[38]. We found that c-di-GMP induced IRF1-dependent Casp1 and Gsdmd mRNA and protein expression in DCs (Fig. 5g, h). Irf1$^{-/-}$-deficient DCs expressed significantly less Gasdermin D mRNA expression compared to wild-type DCs, and failed to upregulate Gasdermin D mRNA expression in response to c-di-GMP (Fig. 5g). Tmem173$^{-/-}$ DCs expressed Gasdermin mRNA comparable to wild-type DCs and failed to significantly upregulated Gasdermin D mRNA in response to c-di-GMP (Fig. 5g). Casp1 expression was significantly reduced in Tmem173$^{-/-}$ and Irf1$^{-/-}$ DCs (Fig. 5g). We next analyzed expression of full-length and cleaved Gasdermin D and Caspase-1 in wild-type, Tmem173$^{-/-}$, and Irf1$^{-/-}$ DCs in response to c-di-GMP stimulation. Caspase-1 and Gasdermin D protein levels increased upon STING activation over an 18-h time period (Fig. 5h). Irf1$^{-/-}$ DCs were unable to respond to stimulation with c-di-GMP with the induction of Gasdermin D and Caspase-1 (Fig. 5h). Remarkably, Tmem173$^{-/-}$ DCs demonstrated elevated Gasdermin D and Caspase-1 protein expression independent of c-di-GMP stimulation (Fig. 5h).

These experiments identified IRF1 as a critical transcription factor that was responsible for the induction of IL-1 family member cytokines and Gasdermin D in the STING pathways. IRF1-mediated transcription was further required as an upstream signal that regulated enzymes of the prostaglandin synthesis pathway and controlled IL-12 and IL-23 receptor components in DCs.

**IRF1 controlled IL-1 expression is required for T$_H$17 cell induction.** We aimed next to understand whether the lack of IL-1β or eicosanoid synthesis was responsible for the lack of T$_H$17 cell generation during antigen presentation by Irf1$^{-/-}$ DCs. We stimulated wild-type or Irf1$^{-/-}$ DCs with IL-1α or IL-1β 18 h after uptake of OVA in the presence of c-di-GMP, and determined the resulting T cells polarization during subsequent antigen presentation (Fig. 6a). Both IL-1α and IL-1β were able to re-establish IL-17A as well as IFN-γ secretion by T cells during antigen-specific activation by Irf1$^{-/-}$ DCs (Fig. 6a).

We next determined whether IL-23 or PGE2 alone or in combination with IL-1β controlled T cells polarization during antigen presentation and T cell activation (Fig. 6b, c). We found that the substitution of IL-1β during T cell activation was able to restore T$_H$17 cell generation by Irf1$^{-/-}$ DCs to the same levels observed in wild-type DC-T cell co-cultures (Fig. 6b). Remarkably, the combination of IL-1β and PGE2 induced significantly higher IL-17A production from T cells compared to IL-1β alone during antigen presentation by both wild-type and Irf1$^{-/-}$ DCs

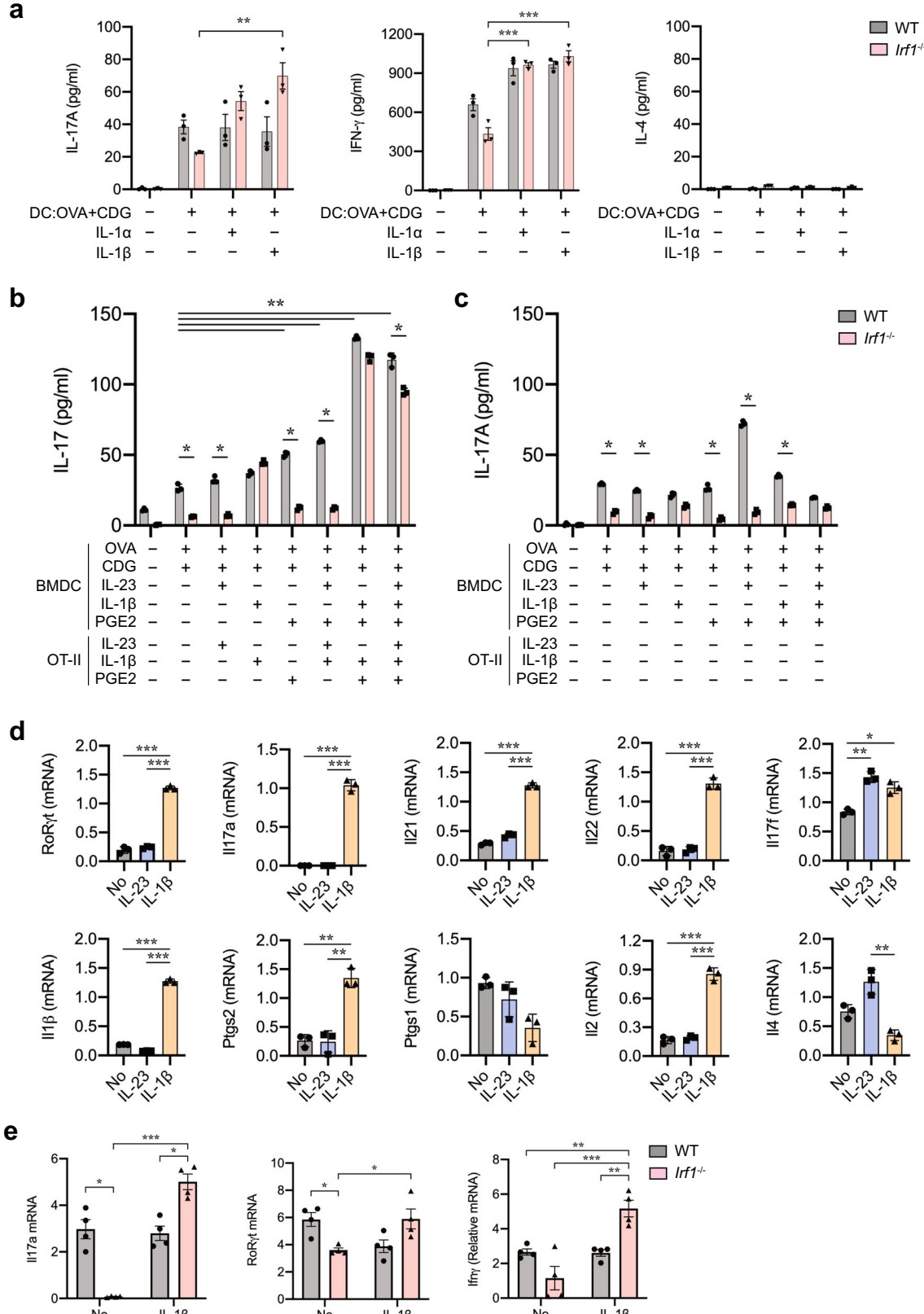

(Fig. 6b). IL-1β was targeting T cells for $T_H17$ differentiation as stimulating $Irf1^{-/-}$ DCs alone or with a combination of IL-1β, IL-23, and PGE2 before the addition of T cells failed to restore $T_H17$ differentiation (Fig. 6c).

To further define the role of IL-1β and IL-23 on T cells polarization during antigen presentation, we co-cultured with $Irf1^{-/-}$ DCs after OVA uptake with naïve OT-II cells and determined the resulting polarization-associated gene programs after cell sorting (Fig. 6d). These experiments revealed that IL-1β induced significant expression of $RoR\gamma t$, $Il17a$, $Il21$, $Il22$, $Il1\beta$, and $Il2$ and reduced $Il4$ mRNA expression (Fig. 6d). IL-1β and IL-23 both were able to induce $Il17f$ expression, while IL-23

**Fig. 6 IL-1β is necessary for T$_H$17 cell differentiation during antigen presentation. a** BMDCs from wild-type and *Irf1*$^{-/-}$ mice were stimulated with OVA plus c-di-GMP for 18 h and washed three times with medium and co-cultured with MACS-sorted naïve OT-II cells in the presence of 5 ng/ml IL-1α or 5 ng/ml IL-1β. **a** IL-17A and IFN-γ production was measured in supernatants by ELISA at day 4. $n = 4$. **$p < 0.01$, ***$p < 0.001$. **b, c** BMDCs from wild-type and *Irf1*$^{-/-}$ mice were stimulated with OVA plus c-di-GMP in the presence of 20 ng/ml IL-23, 5 ng/ml IL-1β, and/or 10 μM PGE and washed three times with medium and co-cultured with MACS-sorted naïve OT-II cells in the presence **b** or absence **c** of IL-23, IL-1β, and/or PGE. IL-17A production in cell supernatants of BMDC:T cells co-cultured for 4 days was measured by ELISA. $n = 3$. *$p < 0.05$, **$p < 0.001$. **d** IL-23 and IL-1β treated CD4$^+$ T cells from co-cultured *Irf1*$^{-/-}$ BMDCs were sorted at day 4 by flow cytometry. mRNA expression of the indicated genes was assessed by qRT-PCR. $n = 3$. *$p < 0.05$, **$p < 0.01$, ***$p < 0.001$. **e** wild-type and *Irf1*$^{-/-}$ mice were immunized with OVA plus c-di-GMP through i.p. route and recombinant mouse IL-1β were injected on day 1 and day 3 and sacrificed on day 5. CD4$^+$ T cells were sorted from MLNs by MACS separation kit. qRT-PCR analysis of *Il17a*, *RoRγc,* and *Ifnγ* expressions in CD4$^+$ T cells from *Irf1*$^{-/-}$ verses wild-type mice. $n = 4$. *$p < 0.05$, **$p < 0.01$, ***$p < 0.001$. *p*-values, one-way ANOVA **d** or two-way ANOVA **a**, **b**, **e** followed by Tukey's multiple-comparisons test or unpaired two-tailed Student's *t*-test **a–c**. Data shown represent two independent experiments.

significantly enhanced *Il4* expression (Fig. 6d). IL-1β further specifically induced expression of *Ptgs2*, while inhibiting the *Ptgs1* mRNA expression during antigen-mediated T-cell activation (Fig. 6d).

We also supplemented IL-1β during in vivo immunization studies with OVA plus c-di-GMP in *Tmem173*$^{-/-}$ and *Irf1*$^{-/-}$ mice. Five days after immunization, CD4$^+$ T cell isolated from MLNs expressed significantly less *Il17a* and *Ifnγ* in *Irf1*$^{-/-}$ and *Tmem173*$^{-/-}$ mice (Fig. 6e). IL-1β substitution at day 3 and day 5 after immunization was able to rescue expression of *RoRγt*, *Il17a*, and *Ifnγ* by CD4$^+$ T cells in MLNs of *Irf1*$^{-/-}$ mice (Fig. 6e). Thus, providing IL-1β during antigen-specific activation of naïve T cells in the presence of c-di-GMP in vitro and in vivo re-established RORγt and cytokine expressions responsible for T$_H$17 cell differentiation by *Irf1*$^{-/-}$ DCs. Although IL-1 was able to compensate for the lack of c-di-GMP-induced IRF1 signaling, IL-1 receptor activation of *Tmem173*$^{-/-}$ DCs or T cells alone was unable to replace the entirety of innate immune signals that were activated by c-di-GMP as IL-1 alone was unable to reestablish *Il17* and *Ifnγ* expressions by T cells in STING-deficient mice (Supplementary Fig. 4).

In aggregate, these results indicated that STING-dependent activation of IRF1 facilitated IL-1 expression and subsequent eicosanoid synthesis that together were essential for T$_H$17 differentiation during antigen presentation.

**STING-IRF1 signaling controls T$_H$17 cells during mucosal host defense**. We next resolved whether STING signaling was required for the induction of T$_H$17 cells in response to infection with the entero-invasive pathogens *Salmonella enterica* Serovar Typhimurium that is known to induce T$_H$17 cells in the intestinal immune system[39]. We orally infected mice with *S. Typhimurium* 20 h after streptomycin treatment to induce colitis[40], with $2 \times 10^8$ C.F.U. in 200 μl suspension in or treated with sterile PBS as control. CD11c$^+$ DCs and CD4$^+$ T cells were isolated 2 days after infection from MLNs to determine DCs activation, T$_H$17 polarization, and *Ifnγ* mRNA expression by T cells.

We found that uptake and transport of *S. typhimurium* to MLNs was significantly enhanced in *Irf1*$^{-/-}$ mice but not impaired in *Tmem173*$^{-/-}$ mice compared to wild-type mice (Fig. 7a). Remarkably, in the presence of *S. typhimurium*, DCs isolated from MLNs from *Tmem173*$^{-/-}$ and *Irf1*$^{-/-}$ mice expressed significantly less *Il1α*, *Il1β*, *Casp1*, *Ptges1,* and *Ptgs2* indicating that the STING–IRF1 signaling axis was critical to mucosal defense activation (Fig. 7b). STING or IRF1-deficient DCs isolated from MLNs during *S. typhimurium* infection also expressed significantly less Gasdermin D, while Caspase 1 expression was found to be IRF1 dependent (Fig. 7b). The impaired DCs function in *Tmem173*$^{-/-}$ and *Irf1*$^{-/-}$ mice resulted in a significant lack of *Il17a*, *Il22*, and *Ifnγ* expression by CD4$^+$ T cells isolated from MLNs from these mice compared to wild-type mice (Fig. 7c).

Together these experiments demonstrated that the STING–IRF1-signaling axis had a pivotal function for the control of DCs that initiated T$_H$17 cell responses during host defense activation by *S. Typhimurium*.

## Discussion

These studies identified microbial CDNs as an important signal in the mucosal immune system for the differentiation of mucosal T$_H$17 cells. CDNs are synthesized in bacteria, where they control virulence, stress survival, motility, antibiotic production, metabolism, biofilm formation, and differentiation[41]. Genes encoding enzymes involved in synthesis and degradation of CDNs are recognizable in the genomes of Gram-positive and Gram-negative bacterial species[41]. CDNs were able to activate IRF1, IRF3, STAT1, NF-κB, and c-Jun all of which could shape the mucosal immune response. We found *Tmem173*$^{-/-}$ DCs unable to induce T$_H$17 cells as they lacked the expression of cytokines, such as *Il6*, *Il23a*, *Il12a*, and *Il27*. Mucosal DCs that expressed the transcription factor Zbtb46 facilitated development of T$_H$17 cells during antigen presentation in the presence of c-di-GMPs, as well as mucosal MHC II-positive CX3CR1$^+$CD11b$^+$CD103$^-$ macrophage-derived DCs.

We identified IRF1 as the critical transcription factors activated by STING signaling that was responsible for the generation of T$_H$17 cells during antigen presentation. We showed that the two most common human STING variants that are able to detect c-di-GMP could form immune complexes with IRF1. Further, STING expression enhanced serine/threonine phosphorylation of IRF1. However, ABs specific for phosphorylated residues of IRF1 will be required to assess IRF1 activation in the STING pathway more precisely in vivo. Further, the pathway-specific kinases that phosphorylated IRF1 upon STING signaling are unknown and may differ from those activating IRF1 during TLR or cytokine receptor signaling. Single nucleotide polymorphisms of human STING define interactions with CDNs and it will be important to determine whether genetic variants in *Tmem173* impact IRF1 activation and subsequently alter mucosal innate and adaptive immunity.

IRF1 expression in DCs was responsible for driving T cell activation, as IRF1 sufficient T cells failed to proliferate and acquire a T$_H$17 cell phenotype in *Irf1*$^{-/-}$ mice. In contrast, IRF1 expression in T cells was not required for T$_H$17 cell differentiation initiated by anti-CD3-stimulated TCR signaling under T$_H$17 polarizing conditions[42].

IL-1β substitution allowed *Irf1*$^{-/-}$ DCs to direct naïve CD4$^+$ T cells to T$_H$17 lineage commitment characterized by the expression of *RoRγt*, *Il17a*, *Il21*, *Il22*, and *Il31* during antigen presentation in the presence of c-di-GMP in vitro and in vivo. A role of IL-1 in the generation of T$_H$17 cells is further supported by the observation that mice deficient in IL-1RI failed to induce IL-17 upon antigen challenge and that IL-23 failed to sustain IL-17 in *Il1r1*-deficient T cells[43]. The combination of IL-23 plus either

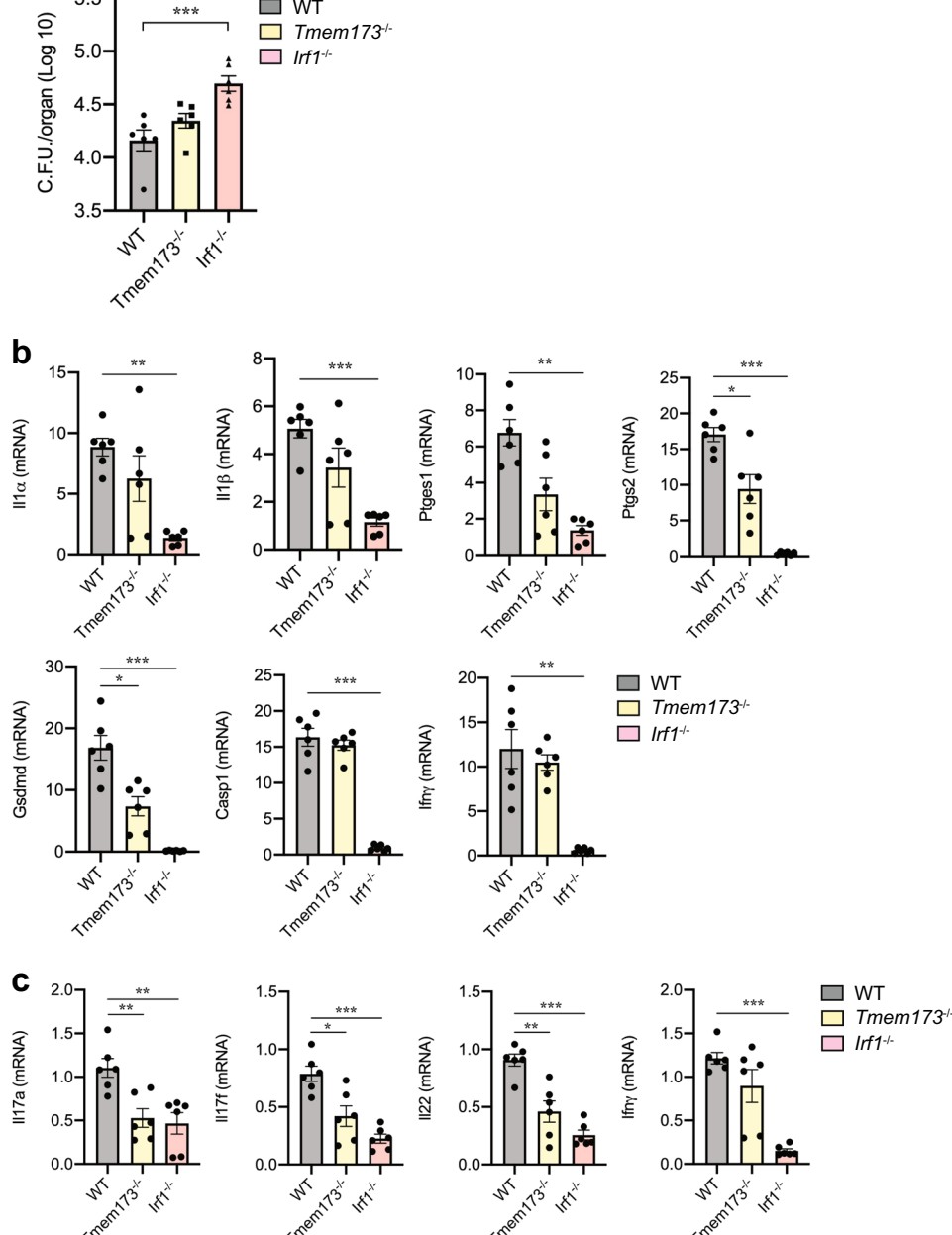

**Fig. 7 The Sting–IRF1 signaling axis controls T$_H$17 differentiation during *S. typhimurium* infection. a–c** Mice were infected with $2 \times 10^8$ C.F.U. of *S. typhimurium* via oral gavage after 20 h of streptomycin treatment. Two days later, MLN were isolated to detect bacteria burden and CD11c$^+$ DCs and CD4$^+$ T cells sorted by MACS to identify gene expression. **a** Numbers of colony detected in MLNs on day 2 after *S. typhimurium* infection. Mean ± SEM. ***$p < 0.001$ by one-way ANOVA. **b** qRT-PCR analysis of *Il1α*, *Il1β*, *Ptgs2*, *Ptges1*, *Ifnγ*, *Casp1*, and *Gsdmd* gene expressions in CD11c$^+$ DCs from MLNs of wild-type, *Tmem173$^{-/-}$*, and *Irf1$^{-/-}$* mice. Mean ± SEM. *$p < 0.05$, **$p < 0.01$, ***$p < 0.001$ by one-way ANOVA. **c** qRT-PCR analysis of *Il17a*, *Il17f*, *Il22*, and *Ifnγ* expressions in MACS-sorted MLN CD4$^+$ T cells from wild-type, *Tmem173$^{-/-}$*, and *Irf1$^{-/-}$* mice. $n = 6$, Mean ± SEM. *$p < 0.05$, **$p < 0.01$, ***$p < 0.001$ by one-way ANOVA. Data shown represent two independent experiments.

IL-1α or IL-1β is synergistic in the induction of IL-17A[44]. Furthermore, mice deficient in both IL-1α and IL-1β do not develop experimental autoimmune encephalomyelitis (EAE), which is caused by pathogenic T$_H$17 cells[45]. As further requirement for T$_H$17 cell development, c-di-GMP induced IRF1-dependent caspase-1 and Gasdermin D expressions, both of which are responsible for the processing and secretion of IL-1β. *Tmem173*-deficient DCs expressed elevated levels of Gasdermin D and Caspase-1 suggesting that the lack of IL-1 expression was responsible for the inability of *Tmem173$^{-/-}$* DCs to induce T$_H$17 cell differentiation.

Our data indicate that additional STING-dependent signals are involved in T$_H$17 cell induction in response to microbial c-di-GMP as IL1-receptor signaling alone was not able to completely re-establish *Il17* and *Ifnγ* expressions by T cells in *Tmem173$^{-/-}$*-deficient mice. We found that STING signaling utilized IRF1 and IRF3 to control expression of Prostaglandin E Synthase (Ptges), Prostaglandin-Endoperoxide Synthase 2 (Ptgs2) linking the impaired antigen-specific T cell responses in *Irf1$^{-/-}$* mice to a compromised activation of the eicosanoid system. Prostaglandin E2 has been shown to promote differentiation and proinflammatory functions of human and murine

$T_H17$ cells through EP2-mediated and EP4-mediated signaling[46]. Further, prostaglandins up-regulated *Il23* and *Il1* receptor expressions, and synergized with IL-1 and IL-23 in the generation of $T_H17$ cells[46].

We further found evidence that STING-induced IRF3/7 activation was specifically required for the induction of *Il22*, *Il34*, *Il9*, *Il33*, and *Ifna4*. Furthermore, *Irf3/7*[−/−] DC expressed significantly increased amount of IFN-γ compared to wild-type and *Irf1*[−/−] DCs. The sustained expression of IFN-γ may contribute to impaired $T_H17$ cell differentiation by *Irf3/7*[−/−] DCs as interferon signaling can prevent $T_H17$ differentiation[47]. In agreement, T cell induced by *Irf3/7*[−/−] DCs still secreted IFN-γ comparable to T cell induced by wild-type DCs. Also, the observed increased *Irf8* expression in *Irf3/7*[−/−] DCs may impair $T_H17$ differentiation as *Irf8*[−/−] mice show increased $T_H17$ lineage commitment[48].

*S. typhimurium* is a intracellular pathogen that causes serious gastrointestinal and systemic infections with the induction of $T_H17$ cells[49,50]. $T_H17$ cell generation during mucosal host defense responses to *S. typhimurium* was dependent on IRF1, indicating that IRF1-induced DC effectors are master regulator for the differentiation of $T_H17$ T cells. The induction of $T_H17$ response to intestinal *S. typhimurium* infection also requires Myd88 and IL-1R-mediated responses, consistent with requirement of IL-1 family cytokine during antigen-specific $T_H17$ cell differentiation[51]. In addition to synthesizing CDNs, *S. typhimurium* can activate several microbial pattern recognition receptors through lipopolysaccharides and bacterial effectors that contribute to innate immune activation[52]. Nevertheless, STING signaling contributed significantly to IL-1 and Gasdermin D expressions by mucosal DCs during *S. typhimurium* infection. The importance of STING-mediated IRF1 activation for host defense against *S. typhimurium* was also apparent in the significant lack of $T_H17$ cell induction in the draining of MLNs of infected mice. $T_H17$ cells have been shown to play a pivotal role in protective immunity by preventing the dissemination of *S. typhimurium* to the MLN[39] and play an important role in resistance to mucosal *Salmonella* infections[50,53]. Patients with a genetic deficiency in $T_H17$ development are also highly susceptible to disseminated *Salmonella* infections[53]. It will need to be determined, whether STING variants that differ in the ability to recognize microbial CDNs cause distinct susceptibility to *Salmonella* infection by inhibiting the induction of protective $T_H17$ cells in the intestine.

Altogether, we identified IRF1 as a pivotal transcription factor in the STING nucleotide-sensing pathway responsible for defining the outcome of antigen presentation by mucosal DCs. STING signaling was not limited to the activation of type-I interferon responses through IRF3, but included a IRF1-dependent transcriptional program that endowed DCs with the ability to drive $T_H17$ polarization in the intestine. IRF1 is a critical component of CDN recognition that allows mucosal DCs to induce host defense through $T_H17$ cells and shape protective immune responses in the intestine to cope with entero-invasive pathogens such as *S. typhimurium*. The identification of a STING–IRF1-signaling axis for adaptive host defense control will aid in the elucidation of disease mechanisms that are initiated by cytosolic host or pathogen DNA processing and recognition.

## Methods

**Mice**. C57BL/6 wild-type (Wild-type), *Il17a*-GFP, *Irf1*[−/−], *Cx3cr1*-GFP, *Zbtb46*-GFP, CAG::KikGR, and OT-II TCR transgenic animals were obtained from Jackson Laboratory (Bar harbor, ME). *Il17a*-GFP mice and CAG;;KikGR mice were crossed with *Tmem173*[−/−] mice. *Tmem173*[−/−] mice (kindly provided by Dr. Glen N. Barber), *Irf3/7* double knockout mice (kindly provided Dr. Michael S. Diamond)[54] (42), *Irf1*[−/−] mice (kindly provided by Dr. Tadatsugu Taniguchi), and *Irf3/Ifnar* double-knockout mice (kindly provided by Dr. Nir Hacohen). All animals were bred and housed in a pathogen-free or in Helicobacter/Pasteurella pneumotropica (HPP)-free animal facility according to institutional guidelines. All experiments were carried out on sex-matched mice at 6–12 weeks old with protocols approved by the subcommittee on Research Animal Care at the Massachusetts General Hospital.

**Generation of BMDCs and co-culture with OT-II naïve CD4$^+$ T cells**. Bone marrow-derived DCs (BMDC) were generated by plating bone marrow cells freshly isolated from tibia and femur into 10 cm dishes and cultured with RPMI-1640 supplemented with 10% heat-inactivated FBS, 1% penicillin and streptomycin, mGM-CSF (10 ng/ml; eBioscience) and mIL-4 (10 ng/ml; eBioscience). On day 6, floating and loosely attached cells were collected representing the BMDCs. For co-culture with CD4$^+$ T cells: BMDCs ($2 \times 10^4$ cells/well) were seeded in 96-well round-bottomed plate and pulsed either with OVA alone (20 μg/ml) or OVA + c-di-GMP (25 μg/ml) for 18 h. After gentle washing, BMDC were cultured for 4 days with CD4 T cells ($1 \times 10^5$ cells/well) purified (by magnetic selection; Miltenyi Biotec) from spleen of OT-II transgenic mice. Supernatants were analyzed by ELISA. For co-culture substitution experiment: PGE2 (Sigma-Aldrich) and the cytokines mIL-23, mIL-1α, and mIL-1β (Invitrogen) were added to BMDC culture alone or to the co-culture with CD4$^+$T cells. Supernatant was analyzed by ELISA and BMDCs were separated from CD4$^+$ T cells by FACS sorting and total RNA analyzed by qRT-PCR.

**Immunization**. C57BL/6 wild-type mice, *Il17a*-GFP, and *Il17a*-GFP; *Tmem173*[−/−] mice were given three times intranasally 20 μg of OVA and 25 μg of c-di-GMP with 2-week intervals between each immunization. Seven days after the third immunization, mice were euthanized, CLNs and MLNs were harvested and single cell suspension was generated using 70 μm cell strainer. A total of $2 \times 10^5$ cells were cultured in 96-well plate and pulsed with 20 μg/ml OVA for 48 h. IFN-γ and IL-17A contained in supernatant were quantified by ELISA assay.

**Surgery and KikGR photoconversion**. KikGR mice were anesthetized, abdominal hair was removed and small incision was made at the center of upper abdomen. Approximately 3 in. of small intestine was gently pulled out and placed on the operating table, and then exposed with 405 nm violet laser (350 mW) for 3 min. The incision was closed with 4–0 silk suture and each mouse received intraperitoneally 200 μg of c-di-GMP. After 18 h, mice were euthanized and the dendritic cell subsets that migrated into MLNs were analyzed using FACS.

**Primary cell isolation**. For T cells or DCs isolation: spleen, Peyer's Patch, and MLN were incubated at 37 °C for 1 h with a solution containing 1 mg/ml Collagenase A (Roche Applied Science) and RPMI medium supplemented with 10% FBS. Five mM EDTA was added and a single cell suspension was generated by passage through a 70 μm cell strainer. Red blood cells were lysed by ACK lysis buffer (Gibco). To isolate DCs from lamina propria, the whole small intestine was cut into three pieces, inverted on polyethylene tubes (Becton Dickenson, Franklin Lakes, NJ), washed three times with PBS 1X and treated with 1 mM dithiothreitol (DTT) to remove mucus. The intestinal epithelium was removed by treatment with 30 mM EDTA/EGTA and digested for 2 h at 37 °C and 5% CO$_2$, with a solution containing 0.125 mg/ml Liberase TL (Roche), complete RPMI medium and 50 μg/ml DNase I (Roche). The digested tissues were passed through a 70 μm cell strainer to generate single cell suspension. DCs were enriched by density centrifugation at 2000 rpm for 20 min at 4 °C using OptiPrep (Axis shield) or Percoll (44% and 67%). The purified DCs and T cells were stained and analyzed by flow cytometry.

**Real-time quantitative-PCR**. Total RNA was isolated using RNeasy micro-kit (Qiagen). cDNA was prepared from RNA using iScript cDNA synthesis kit (Bio-Rad). Real-time PCR was performed using SsoAdvancedTM Universal SYBR green Supermix (Bio-Rad). The gene expression was normalized to the expression of the gene encoding GAPDH or 18S. The primer sequences are provided in Supplementary Table 1.

**Measurement of cytokine production**. IFN-γ, IL-1α, IL-1β, and IL-17A were quantified from cell culture supernatant using standard sandwich ELISA kits (Invitrogen) according to the manufacturer's instructions.

**Flow cytometry and cell sorting**. Flow cytometry was performed on BMDCs or cells isolated from spleen, lymph nodes, or lamina propria. Single cell suspensions were washed with PBS, blocked with Fc receptor-blocking (93; 1:100; Biolegend) and then stained for cell surface antigens using the following anti-murine antibodies conjugated to fluorophore for 20 min at 4 °C: CD103 (M290, 1:100), CD11c (HL3, 1:100), CD3 (145-2C11, 1:100), CD4 (GK1.5, 1:100), MHC II (M5/114.15.2, 1:100), CD11b (M1/70, 1:100), or CD45 (30-F11, 1:100). Washing and antibody incubations were performed in FACS buffer (PBS, 2% FBS). To exclude the dead cells during the analysis, 4′,6-diamidino-2-phenylindole (DAPI; Invitrogen) or 7-amino-actinomycin D (7AAD; BD Biosciences) was added to the cells and directly acquired on BD LSR II flow cytometer or FACS Aria II (BD Bioscience). Data was analyzed using FlowJo software (Tree Star).

**Antibodies and Western blotting**. The following antibodies used in this study were purchased from Cell Signaling: IRF1 (D4E4), STING (D2P2F), Lamin A/C (4C11), Phospho-c-Jun (D47G9), c-Jun (60A8), Phospho-IRF3 (4D4G), IRF3 (D83B9), Phospho-p65-NFκB (93H1), p65-NFκB (D14E12), Phospho-STAT1 (58D6), STAT1 (cat no. 9172), IL-1α (D4F3S), Phosphoserine/threonine (cat no. 9631), HA-Tag (C29F4), and anti-β-actin (8H10D10). Whole-cell were lysed with lysis buffer (1% NP-40, 20 mM Tris–HCl (pH 7.4), 150 mM NaCl, 2 mM EDTA, 2 mM EGTA, 4 mM Na$_3$VO$_4$, and 40 mM NaF) containing protease and phosphatase inhibitors tablets (Roche).

Western blotting was performed according to standard procedures for SDS–PAGE and wet transfer onto PVDF membranes. Membranes were blocked with blocking buffer (BSA 5% in TBST) at room temperature for 1 h and incubated with primary antibodies overnight at 4 °C. Secondary anti-mouse (NA931V, GE Healthcare) or rabbit (NA934V, GE Healthcare) HRP were used at 1/5000 and incubated for 1 h at RT. The bands were visualized by enhanced chemiluminescence (Western Lightning Plus [PerkinElmer] or SuperSignal West Femto [Thermo Fisher Scientific]) and exposure on X-ray Film Processor SRX-101 (Konica Minota). The blots probed for the phosphorylated proteins were stripped and re-probed with antibodies for the respective total proteins.

**Immunoprecipitation and subcellular fractionation**. pUNO1-human IRF1-HA plasmid was purchased from Invivogen and pcDNA3-human STING (R232H)-HA vector was a gift from Dr. Glen N. Barber. STING (R232R)-HA variant was generated from pcDNA3-huSTING construct using the QuikChange II Site-directed mutagenesis Kit (Stratagene) according to the manufacturer's instructions. HEK293T cells were transfected with indicated plasmids using Lipofectamine 3000 (Invitrogen). The cells were lysed on ice for 20 min, in the same lysis buffer used to harvest cell lysate for immunoblot. The supernatant collected after centrifugation was incubated for 40 min at 4 °C with protein G plus agarose (Pierce Thermo Scientific, Rockford, IL, USA). The precleared lysates were incubated then with immunoprecipitation antibodies at 4 °C overnight. The protein G agarose beads were added and incubated for 4 h. After extensive washes with lysis buffer, the agarose beads were mixed with 1 × SDS sample buffer and boiled at 95 °C for 5 min and analyzed by western blotting.

Nuclear extracts from BMDCs were prepared using Buffer A (10 mM HEPES, pH 7.9; 1.5 mM MgCl$_2$; 10 mM KCl; 0.1 mM EDTA; 0.1 mM EGTA; 1 mM DTT; 0.3 mM Na3VO4 + protease inhibitors tablet [Roche]) and Buffer C (20 mM HEPES, pH 7.9; 1.5 mM MgCl$_2$; 1 mM EDTA; 1 mM EGTA; 1 mM DTT; 0.3 mM Na$_3$VO$_4$; 0.4 M NaCl + protease inhibitors tablet [Roche]). Cells were collected by scraping and centrifugation, 400 μl of Buffer A added to the pellet and after incubation for 15 min on ice, 50 μl of 10% NP-40 was added and the supernatant was collected (cytosolic fraction) after centrifugation. Volume of 50 μl of Buffer C was added to the pellet, vigorously rocked at 4 °C for 15 min and centrifuged for 5 min at 15,000 rpm. the nuclear extracts obtained from the collected supernatant. Nuclear proteins were boiled with SDS–PAGE sample buffer at 95 °C for 5 min and detected by western blotting using anti-IRF1 antibody or anti-Lamin A/C.

**Inflammasome activation**. BMDCs were plated in 24-well plate at ($1 \times 10^6$ cells/well). We used two different methods to detect IL-1α and IL-1β productions upon c-di-GMP stimulation: In the first, BMDCs were stimulated with 25 μg/ml of c-di-GMP alone for 18 h and in the second, BMDCs were primed for 3 h with 200 ng/ml of LPS and stimulated for another 6 h either with 25 μg/ml or 50 μg/mg of c-di-GMP in the presence of 5 mM of ATP. Supernatant in the first method was collected after 18 h stimulation and in the second method after 6 h stimulation.

**RNAseq**. Total RNA was isolated from C57BL/6 wild-type, $Tmem173^{-/-}$, $Irf1^{-/-}$, $Irf3/7^{-/-}$, and $Irf7^{-/-}$ DCs using RNeasy Micro-kit (Qiagen) following the manufacturer's instructions. Libraries were synthesized using Illumina TruSeq-stranded mRNA sample preparation kit from 500 ng of purified total RNA and indexed adaptors according to the manufacturer's protocol (Illumina). The final dsDNA libraries were quantified by Qubit fluorometer, Agilent Tapestation 2200, and qRT-PCR using the Kapa Biosystems library quantification kit according to manufacturer's protocols. Pooled libraries were subjected to 35-bp paired-end sequencing according to the manufacturer's protocol (Illumina NextSeq 500). Targeted sequencing depth was 25 million paired end reads per sample. Blc2fastq2 Conversion software (Illumina) was used to generate de-multiplexed Fastq files. Expression values were normalized as Fragments per Kilobase Million reads after correction for gene length (FPKM) in Cuffdiff version 1.05 in the DNAnexus analysis pipeline and filtered for genes that exhibited a statistically significant difference ($p < 0.01$) with an FDR threshold of 0.05 and a biologically relevant change log fold change >1. Samples were analyzed in the RNA-sequencing pipeline of Seqmonk for mRNAs for opposing strand specific and paired end libraries with merged transcriptome isoforms, correction for DNA contamination and log transformed resulting expression values in log$_2$ FPKM.

**Salmonella infection**. Mice were fasted for 4 h prior to 20 mg of streptomycin treatment. Twenty hours after streptomycin treatment, mice were fasted for additional 4 h before the oral administration of 200 μl PBS containing $2 \times 10^8$ C.F.U. of S. Typhimurium. After the infection, drinking water was given immediately and

food was provided 2 h later. Two days later, mice were euthanized, and single cell suspensions were prepared from MLNs to purify CD11c$^+$ DCs population and CD4$^+$ T cells using magnetic separation. Total RNA was extracted from the purified cells and gene expression was quantified by qRT-PCR assay.

In order to monitor the intracellular Salmonella population, MLNs from inoculated mice were lysed in 0.5% Triton X-100 and adequate dilutions were plated on 50 μg/ml streptomycin MacConkey Agar plates to determine the C.F.U.

**Statistics and reproducibility**. All data are presented as mean ± S.E.M. except when indicated otherwise. $n$ represents the number of animals per experiment. Statistical significance was calculated by two-tailed unpaired Student's $t$-test, one-way ANOVA and two-way ANOVA with Tukey's test. All statistical tests were performed with GraphPad Prism Software (version 8.12; GraphPad, San Diego, CA, USA). Differences of $p < 0.05$ were considered statistically significant.

**Reporting summary**. Further information on research design is available in the Nature Research Reporting Summary linked to this article.

## Data availability
The RNA-seq data have been deposited to the Gene Expression Omnibus under the accession no. GSE137428. Available at https://www.ncbi.nlm.nih.gov/geo/query/acc.cgi?acc=GSE137428. Other source data that support the findings of this study are available in the supplementary materials or from the corresponding author upon request.

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

### Acknowledgements

This work was supported by grants DK068181 (H.-C.R.), AI113333 (H.-C.R.), and DK043351 (H.-C.R.) from the National Institutes of Health.

### Author contributions

H.-C.R. designed experiments; and S.-M.P, T.O., Y.Z., N.Y., R.Z. and P.S. carried out experiments; S.-M.P., T.O., R.Z. and H.-C.R. analyzed and interpreted data and S.-M.P, R.Z. and H.-C.R. prepared the manuscript.

### Competing interests

The authors declare no competing interests.
