## [Peer Review File · Communications Biology]

Reviewers' comments:

Reviewer #1 (Remarks to the Author):

The study by Park, Reinecker and et al. describes an interesting in vivo axis of the STING-mediated polarization of Th17 cells in response to microbial c-di-GMP. Overall, the manuscript is easy to follow and contains several novel findings. Firstly, while cGAS-STING signaling has been shown to induce IRF1 via type I interferon (Nat Immunol. 2015 May;16 5:467-75.), a direct STING-mediated IRF1 activation appears to be a new discovery. Secondly, while a previous report has documented an impact of IRF1 deficiency on IL-1b expression (J Immunol. 2018 Feb 15;200 4:1489-1495.), the current study is the first to link IL-1b and GSDMD in a functional axis. Thirdly, the study demonstrated a pivotal role of IL-1b in the polarization of mucosal Th17, independent of IL-23 and IL-6. A major weakness of this study is its heavy reliance on ex vivo system. In vivo data presented herein are not sufficient to demonstrate this axis or its role in regulating mucosal Th17. In addition, the interpretation of several data deserves more careful consideration, especially regarding the novel claim on the STING-mediated activation of IRF1. Specific comments are listed below.

1. c-di-GMP clearly induced IRF1 as its protein level elevated in the whole cell lysates (Fig.2C). However, it does not necessary follow that, in addition to induction of IRF1, STING-mediated signaling also directly activates IRF1 as the authors claim. Does co-expression of STING and IRF1 promotes IRF1 to translocate into the nuclei? Can c-di-GMP trigger IRF1 activation in cells that have non-inducible and constitutive IRF1 level?

2. The authors said the expression of GSDMD was "partially dependent on STING (Fig.5h)". It would appear in this case that STING deficient cells had much higher level of GSDMD, although CDG was able to clearly induce its the expression. Notably, this elevated level of GSDMD expression even at baseline correlated with increased IRF1 nuclear translocation in the absence of stimulation as shown in Fig. 2c in STING deficient cells. These inconsistencies undermine the authors' conclusion on the STING-IRF1 signaling.

3. The induction of pro-IL-1b shown in Fig.5f appears to be very weak. Clearly LPS was a robust inducer of pro-IL-1b. Yet CDG did not further increase its level. In fact, CDG also had limited induction of pro-IL-1b transcripts as shown in Fig. 5c.

On a side note, it is unclear what the y axes represent, fold induction or relative level.

4. There are insufficient data to suggest that "the STING-IRF1 signaling axis controls TH17 cell differentiation for mucosal defense against entero-invasive pathogens". This claim implies this axis is crucial for protection against pathogen, which the authors did not demonstrate.

Reviewer #2 (Remarks to the Author):

Park et al. set out to investigate the STING-dependent recognition of c-di-GMP during Salmonella infection and how this activates mucosal dendritic cells important for TH17 cell polarization. The authors show that IRF1 is activated in BMDCs and alters their transcriptional signature. Subsequent induction of IL-1 and gasdermin D seems to be a prerequisite for TH17 polarization. The study is well designed and combines in vivo and in vitro approaches where necessary. Knowledge on the STING-IRF1 axis following c-di-GMP exposure is limited and therefore this manuscript shows novel and exciting findings in the context of pathogen recognition and priming of mucosal adaptive immune responses.

The manuscript would highly benefit from a rewriting of the introduction and discussion as the message and the meaning of the results in context of the literature are not always clear to the reader. Please introduce the role of TH17 cells following Salmonella infection and what is known about c-di-GMP in this context. Additionally, an introduction of IRF1 would be helpful to put the obtained data into context. The discussion fails to provide a bigger picture, which pathways are utilized upstream and downstream of IRF1 (dependent and independent of STING) and how this might influence the outcomes of the infection.

Technical/minor comments:

Fig. 1

1a: Please indicate which groups were compared for the statistical analysis.

1c: It seem as the migration of both DC subsets differs. Please additionally show % increase as the starting frequency is different.

1e: Both DC subsets induce IL-17 production in OT-II T cells. Please additionally show % increase as well and comment in text.

Fig. 2 Some of the blots show very weak bands and it is recommended to repeat these experiments.

Fig. 3 The depicted in vivo experiments use *Irf1*^{-/-} mice. The results obtained using these mice can be independent of IRF1-expression in DCs. Are DC-specific *Irf1* KO mice available? Also, it would be interesting to repeat Fig. 3d and 3e with *Tmem*^{-/-} mice – would they show the same outcome?

Fig. 4 I recommend focussing on the comparison between *Irf1*^{-/-} and *Tmem*^{-/-} gene expression signature to validate the general hypothesis of the manuscript. Especially a focus on TH17-inducing genes would be helpful.

Fig. 6 Pleas repeat experiments with *Tmem*^{-/-} mice to validate the importance of the STING-IRF1 axis, especially the in vivo rescue experiments in Fig. 6e.

Fig.7 Analysis of a CD4 T cell response 2 days post infection is quite early as they require time for priming and proliferation in vivo. STING doesn't seem to be required and another pathway upstream of IRF1 seems to be more important for this in vivo finding. Please comment on this in the discussion.

We thank the reviewers for the insight full review and suggestions that allowed us to improve the manuscript.

Reviewers' comments:

Reviewer #1 (Remarks to the Author):

The study by Park, Reinecker and et al. describes an interesting *in vivo* axis of the STING-mediated polarization of Th17 cells in response to microbial c-di-GMP. Overall, the manuscript is easy to follow and contains several novel findings. Firstly, while cGAS-STING signaling has been shown to induce IRF1 via type I interferon (Nat Immunol. 2015 May;16 5:467-75.), a direct STING-mediated IRF1 activation appears to be a new discovery. Secondly, while a previous report has documented an impact of IRF1 deficiency on IL-1b expression (J Immunol. 2018 Feb 15;200 4:1489-1495.), the current study is the first to link IL-1b and GSDMD in a functional axis. Thirdly, the study demonstrated a pivotal role of IL-1b in the polarization of mucosal Th17, independent of IL-23 and IL-6. A major weakness of this study is its heavy reliance on *ex vivo* system. *In vivo* data presented herein are not sufficient to demonstrate this axis or its role in regulating mucosal Th17. In addition, the interpretation of several data deserves more careful consideration, especially regarding the novel claim on the STING-mediated activation of IRF1. Specific comments are listed below.

Answer: The data that demonstrate the relevance of the STING and IRF1 signaling axis has been established *in vivo* experiment in Fig. 3f. As STING signaling involves the activation of numerous transcription factors in different DC subsets, we had to initially simplify the experimental system to allow the identification of IRF specific gene signature in response to c-di-GMP stimulation (Fig. 3g, Fig 4d, 4e, and 4f, Fig. 5e, Fig. S1a and Fig. S1b). Also, the limited availability of tools to study IRF1 activation required *in vitro* experimentation to confirm STING variant - IRF1 interaction and phosphorylation observed in primary cells. Fig. 2e and 2f).

The key mechanisms controlled by IRF1 and the relevance for T_H17 polarization were finally confirmed by *in vivo* experiments (Fig. 3e, Fig. 3f and Fig 6e) and also confirmed by *in vivo* experiments with entero-invasive *Salmonella typhimurium* infection (Fig. 7).

Comment 1. c-di-GMP clearly induced IRF1 as its protein level elevated in the whole cell lysates (Fig.2C). However, it does not necessary follow that, in addition to induction of IRF1, STING-mediated signaling also directly activates IRF1 as the authors claim. Does co-expression of STING and IRF1 promotes IRF1 to translocate into the nuclei? Can c-di-GMP trigger IRF1 activation in cells that have non-inducible and constitutive IRF1 level?

Answer: We were able to demonstrate in new experiments that the phosphorylation of IRF1 increased in the presence of STING. (Fig. 2f and below). Phosphorylation of IRFs is recognized as an indication of activation and requirement for nuclear translocation. Unfortunately, there are no ABs available that can detect phosphorylation of IRF1 as for IRF3 or IRF5. To determine whether STING can directly induce IRF1 phosphorylation, we co-transfected HA-STING and HA-IRF1 plasmids into 293 T cells and pulled down immunocomplexes using anti-phosphoserine/threonine antibody and performed Western blotting using anti-HA antibody to detect both STING and IRF1 protein. This method has been recently used to determine IRF1 phosphorylation and activation (PMID: 30854564). We found that in the presence of STING the amount of phosphorylated IRF1 increased compared to IRF1 expression alone (Fig. 2f). In addition, we have replaced 'activation' by 'nuclear translocation' to more precisely represent our findings.

Figure 2f. STING induced phosphorylation of IRF1. Antibodies detecting phosphoserine and/threonine were used to immunoprecipitated IRF1 from HEK293T cells that were co-transfected with HA-STING and HA-IRF1 encoding plasmids. IRF1 and STING were detected using HA antibody.

Comment 2. The authors said the expression of GSDMD was “partially dependent on STING (Fig.5h)”. It would appear in this case that STING deficient cells had much higher level of GSDMD, although CDG was able to clearly induce its the expression. Notably, this elevated level of GSDMD expression even at baseline correlated with increased IRF1 nuclear translocation in the absence of stimulation as shown in Fig. 2c in STING deficient cells. These inconsistencies undermine the authors’ conclusion on the STING-IRF1 signaling.

Answer: We have clarified this point. We found that unstimulated *Irf1*^{-/-} DCs expressed significantly less Gasdermin D mRNA compared to WT DCs and failed to upregulate Gasdermin D mRNA expression in response to c-di-GMP (Fig. 5g). *Tmem173*^{-/-} DCs expressed baseline levels of Gasdermin D mRNA that was comparable to WT and failed to significantly upregulate Gasdermin D mRNA in response to c-di-GMP (Fig. 5g). Gasdermin D is expressed at elevated levels in *Tmem173*^{-/-} DCs compared to WT mice (Fig. 5h) but in the absence of IL-1 cytokine expression no IL-1 can support T_H17 polarization can initiate.

Comment 3. The induction of pro-IL-1b shown in Fig.5f appears to be very weak. Clearly LPS was a robust inducer of pro-IL-1b. Yet CDG did not further increase its level. In fact, CDG also had limited induction of pro-IL-1b transcripts as shown in Fig. 5c.

Answer: IL1 expression in *Tmem173*^{-/-} and *Irf1*^{-/-} DCs response to c-di-GMP is shown on mRNA level and protein level in Fig. 5a, 5b, and 5d. *Il1β* mRNA is significantly increased in WT BMDCs after c-di-GMP stimulation (Fig. 5a and 5b) and is independent on IRF3 and type I IFN signaling. (Fig. 5c). IL-1β protein levels also significantly increased after c-di-GMP stimulation in WT BMDCs (Fig. 5d right). IL-1β mRNA levels and protein levels are impaired in *Irf1*^{-/-} DCs in response to c-di-GMP. Fig 5 f demonstrates that LPS can induce Pro-IL-1β in *Tmem173*^{-/-} and *Irf1*^{-/-} DCs in the absence or presence of c-d-GMP. The data in Fig. 5f is a control for the assessment of IL-1β expression by ELISA shown in Fig. 5E. It shows that LPS is able to induce IL1 expression in the absence of STING, but that IRF1 is required for induction in both pathways. We show that IL-1β is more induced in responses to LPS/CDG than LPS alone in WT BMDCs using ELISA (Fig. 5e right).

Comment 4. On a side note, it is unclear what the y axes represent, fold induction or relative level.

Answer: we clarified that the Y axis shows the relative level normalized to β -actin.

Comment 5. There are insufficient data to suggest that “the STING-IRF1 signaling axis controls TH17 cell differentiation for mucosal defense against entero-invasive pathogens”. This claim implies this axis is crucial for protection against pathogen, which the authors did not demonstrate.

Answer: We agree and have modified this statement. It now reads: The STING-IRF1 signaling axis regulates TH17 cell responses during *S. typhimurium* infection.

Reviewer #2 (Remarks to the Author):

Park et al. set out to investigate the STING-dependent recognition of c-di-GMP during Salmonella infection and how this activates mucosal dendritic cells important for Th17 cell polarization. The authors show that IRF1 is activated in BMDCs and alters their transcriptional signature. Subsequent induction of IL-1 and Gasdermin D seems to be a prerequisite for Th17 polarization.

The study is well designed and combines in vivo and in vitro approaches where necessary. Knowledge on the STING-IRF1 axis following c-di-GMP exposure is limited and therefore this manuscript shows novel and exciting findings in the context of pathogen recognition and priming of mucosal adaptive immune responses.

The manuscript would highly benefit from a rewriting of the introduction and discussion as the message and the meaning of the results in context of the literature are not always clear to the reader. Please introduce the role of TH17 cells following Salmonella infection and what is known about c-di-GMP in this context. Additionally, an introduction of IRF1 would be helpful to put the obtained data into context. The discussion fails to provide a bigger picture, which pathways are utilized upstream and downstream of IRF1 (dependent and independent of STING) and how this might influence the outcomes of the infection.

Answer: We have rewritten the introduction and discussions.

Comment 1. Technical/minor comments:

Fig. 1

1a: Please indicate which groups were compared for the statistical analysis.

Answer: We indicated the statistical analysis in Fig. 1a.

Comment 2. 1c: It seem as the migration of both DC subsets differs. Please additionally show % increase as the starting frequency is different.

Answer: In our experiments, we followed the migration of the two main classical DC subsets which are CD103⁺CD11b⁺ and CD103⁺CD11b⁻. Migrated DCs were composed to $37.38 \pm 4.68\%$ of CD103⁺CD11b⁺ DCs and to $12.35 \pm 1.61\%$ of CD103⁺CD11b⁻ in WT mice (Fig. 2d).

Figure 2d. Migration of DC subsets from the lamina propria to MLN in response to STING signaling. Comparison to the two main classical DC subsets in MLNs after c-di-GMP induce migration from the lamina propria.

Comment 3. 1e: Both DC subsets induce IL-17 production in OT-II T cells. Please additionally show % increase as well and comment in text.

Answer: We corrected this in response to your comment. We found that Zbtb46⁺CD103⁺CD11b⁺ DCs significantly induced 230.5 ± 10.7 % more *Il17a* expression in OT-II transgenic T cells compared to Cx3cr1⁺CD103⁻CD11b⁺ DCs in the presence of OVA and c-di-GMP (Fig. 1e).

Comment 4. Fig. 2 Some of the blots show very weak bands and it is recommended to repeat these experiments.

Answer: We improved the image reproduction and used longer exposed westerns.

Comment 5. Fig. 3 The depicted in vivo experiments use *Irf1*^{-/-} mice. The results obtained using these mice can be independent of IRF1-expression in DCs. Are DC-specific *Irf1* KO mice available? Also, it would be interesting to repeat Fig. 3d and 3e with *Tmem*^{-/-} mice – would they show the same outcome?

Answer: Unfortunately, there are no floxed-IRF1 mice available yet for cell type specific deletions. Our experiments using adoptive T cell transfer in Fig. 3d and 3e show that antigen presentation is required for T_H17 cell polarization during immunizations. In STING deficient mice c-di-GMP cannot provide adjuvant function and IL-17A and IFN γ expression is significantly reduced in *IRF1*^{-/-} and *STING*^{-/-} mice upon OVA/c-di-GMP immunizations as shown in Fig. 3f.

Comment 6. Fig. 4 I recommend focusing on the comparison between *Irf1*^{-/-} and *Tmem*^{-/-} gene expression signature to validate the general hypothesis of the manuscript. Especially a focus on TH17-inducing genes would be helpful.

Answer: Only by carrying out the clustering with WT, STING, IRF1 and IRF3/7 gene expression datasets were we able to distinguish IRF1 from IRF3/7 dependent genes that are specifically regulated by c-di-GMP. As *Tmem173*^{-/-} DC do not respond to c-di-GMP at all a side by side comparison to *Irf1*^{-/-} would not reveal the specific set of T_H17 inducing genes. We moved fig 4g,

4h and 4i to New supplementary figure 3 to focus on the IRF1 dependent gene expression signature in Fig. 4. The raw sequencing data has been submitted to the GEO repository and will be available at: <https://www.ncbi.nlm.nih.gov/geo/query/acc.cgi?acc=GSE137428>

Supplementary fig. 3

New supplementary figure 3. Cluster analysis of WT, IRF1 and IRF3/7 dependent gene expression signatures in response to c-di-GMP stimulation.

Comment 7. Fig. 6 Please repeat experiments with *Tmem*^{-/-} mice to validate the importance of the STING-IRF1 axis, especially the in vivo rescue experiments in Fig. 6e.

Answer: This an important point we added to the discussion. STING deficient DCs are entirely unresponsive to c-di-GMP (Fig. 2 c and 2d), lacking IRF1, IRF3, Stat1, NF-κB, c-Jun activation and do not express cytokines required for T_H17 generation such as *Il6*, *Il23a*, *Il12a* and *Il27* (Fig. 3g). As there is no innate immune response elicited, T_H17 generation by STING deficient DCs is severely impaired (Fig. 3c 3f).

The manuscript is focused on elucidating the specific role of STING induced IRF1 dependent gene programs for T_H17 cell differentiation. We can only compensate for the lack of c-di-GMP induced IRF1 signaling by substituting IL-1. IL-1 receptor activation of *Tmem173*^{-/-} DCs alone cannot replace all the innate immune signals that are activated by c-di-GMP. We carried out *i.p.* immunizations with OVA c-di-GMP in the absence or presence of IL-1β and found that IL1 alone was unable to reestablish full T_H17 generation in *Tmem173*^{-/-} mice (shown below and new supplementary Figure 4).

Supplementary Figure 4. IL-1β injection cannot re-establish IL-17A expression in response to OVA/c-di-GMP immunization in *Tmem173*^{-/-} mice.

Which of the other STING activated pathways is linked to IL6 expression for example will require extensive new investigations. Our data also indicate that additional STING induced signals through IRF3 may further contribute to T_H17 subdifferentiation as expression of *Ahr*, *Dnase1|3* and *Ptger2* are reduced in *Irf1*^{-/-} and *Irf3/7*^{-/-} DCs (Fig. 4f) Also the control of *Ifnγ* by IRF3 may play a role in defining the outcome of antigen presentation in the presence of c-di-GMP.

Comment 8. Fig.7 Analysis of a CD4 T cell response 2 days post infection is quite early as they require time for priming and proliferation in vivo. STING doesn't seem to be required and another pathway upstream of IRF1 seems to be more important for this in vivo finding. Please comment on this in the discussion.

Answer: We followed established protocols (PMID: 17760501). *Salmonella* infection proceeds in mice with the rapid induction of T cell activation. STING deficiency has a limited impact as *Salmonella* derived LPS can also activate IL-1 expression (Fig. 5e-f). In contrast IRF1, is required in both pathways and consequently IRF1 deficient mice are severely impaired in mounting a T_H17 cell response in response to *Salmonella* infection.

REVIEWERS' COMMENTS:

Reviewer #1 (Remarks to the Author):

The authors have satisfactorily addressed my comments. This study conveys a novel finding that is of interest to a growing field. With the authors' efforts to clarify previously noted issues, the manuscript has been substantially improved.

Reviewer #2 (Remarks to the Author):

The authors have done a great job in improving the manuscript. This study demonstrates a role for IRF1 in Th17 differentiation following CDG and Salmonella infection. Therefore, the manuscript would benefit from an introduction/discussion section about Th17 cells in the context of Salmonella infection (Are Th17 cells important? What is their mechanism of action? In which conditions?). Providing the reader with this particular context would highlight the importance and translatability of this study's main findings (as reviewed for example in PMID: PMC3652671).

Minor comments:

Fig 1e: My apologies for not making this point very clear in the first place.

The CD103+CD11b+ DC increase from ~1.5% to ~5.5% following CDG stimulation. This roughly is an increase of 3.6-fold. The CD103+CD11b- DC increase from ~0.5% to ~2%, which represents a 4-fold increase. This demonstrates that both DC subsets analysed show a similarly increased migration following CDG.

Fig. 1g: similar to Fig. 1e.

CX3CR1+ LP-DCs actually increase their IL17a stimulatory capacity more than the Zbtb46+ LP-DC (~3.5 versus ~1.5-fold increase). This doesn't require an extra figure but may be worth mentioning in the text.

Response to the remaining comments by Reviewer 2

The authors have done a great job in improving the manuscript. This study demonstrates a role for IRF1 in Th17 differentiation following CDG and Salmonella infection. Therefore, the manuscript would benefit from an introduction/discussion section about Th17 cells in the context of Salmonella infection (Are Th17 cells important? What is their mechanism of action? In which conditions?). Providing the reader with this particular context would highlight the importance and translatability of this study's main findings (as reviewed for example in PMID: PMC3652671).

Answer: We have expanded the discussion of the impact of our findings on anti-Salmonella Host defense.

Minor comments:

Fig 1e: My apologies for not making this point very clear in the first place. The CD103+CD11b+ DC increase from ~1.5% to ~5.5% following CDG stimulation. This roughly is an increase of 3.6-fold. The CD103+CD11b- DC increase from ~0.5% to ~2%, which represents a 4-fold increase. This demonstrates that both DC subsets analysed show a similarly increased migration following CDG.

Answer: We have included these findings into the paper.

Fig. 1g: similar to Fig. 1e.

CX3CR1+ LP-DCs actually increase their IL17a stimulatory capacity more than the Zbtb46+ LP-DC (~3.5 versus ~1.5-fold increase). This doesn't require an extra figure but may be worth mentioning in the text.

Answer: We have highlighted this relationship in the text.